# Spiking Discrepancy Transformer for Point Cloud Analysis

[1]**Yijie Lu**   [2]**Zhiyi Pan**   [3]**Renrui Zhang**   [4]**Yanhao Jia**   [1]**Ronggang Wang**
[5]**Zhaokun Zhou**[*]
[1]Peking University   [2]Tianjin University   [3]The Chinese University of Hong Kong
[4]Nanyang Technological University   [5]Tencent
luyijie@stu.pku.edu.cn,   zhaokunzhou@tencent.com

## Abstract

Spiking Transformer has sparked growing interest, with the Spiking Self-Attention merging spikes with self-attention to deliver both energy efficiency and competitive performance. However, existing work primarily focuses on 2D visual tasks, and in the domain of 3D point clouds, the disorder and complexity of spatial information, along with the scale of the point clouds, present significant challenges. For point clouds, we introduce spiking discrepancy, measuring differences in spike features to highlight key information, and then construct the Spiking Discrepancy Attention Mechanism (SDAM). SDAM contains two variants: the Spiking Element Discrepancy Attention captures local geometric correlations between central points and neighboring points, while the Spiking Intensity Discrepancy Attention characterizes structural patterns of point clouds based on macroscopic spike statistics. Moreover, we propose a Spatially-Aware Spiking Neuron. Based on these, we construct a hierarchical Spiking Discrepancy Transformer. Experimental results demonstrate that our method achieves state-of-the-art performance within the Spiking Neural Networks and exhibits impressive performance compared to Artificial Neural Networks along with a few parameters and significantly lower theoretical energy consumption.

## 1 Introduction

Spiking Neural Networks (SNNs), regarded as the third generation Neural Networks Maass (1997), are characterized by their biological plausibility Roy et al. (2019), spike-driven characteristics, and low power consumption. By emulating the dynamics of biological neurons, SNNs utilize asynchronous binary spikes for information transmission, with the membrane potential updated only upon the arrival of spikes. This unique feature allows SNNs to avoid unnecessary computations on zero values, making them promising candidates on neuromorphic hardware, such as TrueNorth Merolla et al. (2014) and Loihi Davies et al. (2018).

Researchers are making extensive efforts to enhance the performance of SNNs across various visual tasks, including image classification Fang et al. (2021b;a); Guo et al. (2023); Meng et al. (2023); Xu et al. (2024); Shen et al. (2024b), object detection Su et al. (2023); Luo et al. (2024); Qu et al. (2025), and semantic segmentation Yao et al. (2024a). Recently, inspired by the impressive achievements of vision transformers Dosovitskiy et al. (2021); Liu et al. (2021) in Artificial Neural Networks (ANNs), attempts have been made to incorporate transformer-based architectures into SNNs. Notably, Spikformer Zhou et al. (2023c) introduces Spiking Self-Attention (SSA) mechanism, while the Spike-Driven Transformer Yao et al. (2024b) employs Spike-Driven Self-Attention. Other studies have focused on structural enhancements Yao et al. (2021); Zhou et al. (2023a;b); Zhang et al. (2024b); Qiu et al. (2024b); Deng et al. (2024); Qiu et al. (2025), training methodologies Wang et al. (2023b), and applications across different tasks Bal & Sengupta (2024). However, these efforts are primarily confined to 2D visual domains. The exploration of the Spiking Transformer in 3D point clouds remains limited.

---

[*]Corresponding author.

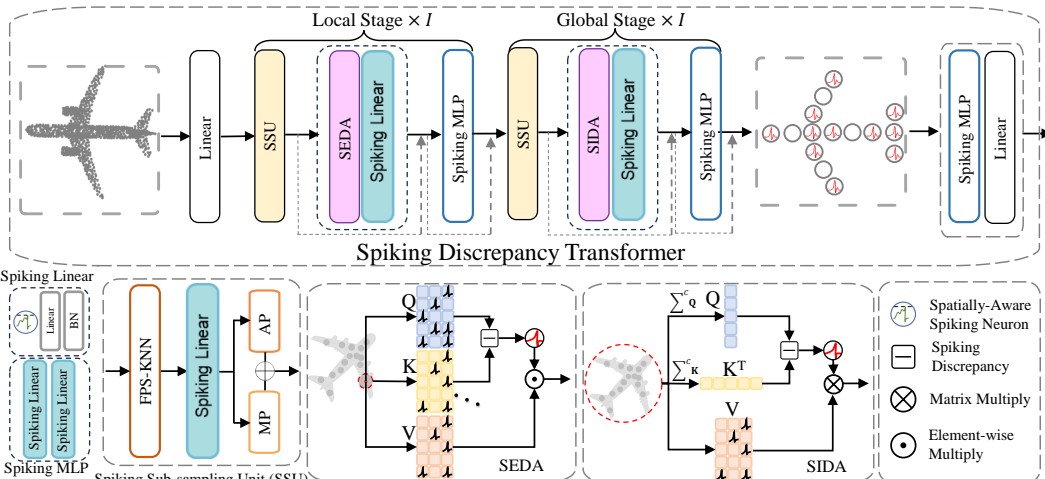

Figure 1: The overview of Spiking Discrepancy Transformer and illustration of key components.

3D point clouds analysis is critical in autonomous driving Chen et al. (2017); Kidono et al. (2011); Navarro-Serment et al. (2010), scene understanding Chen et al. (2019), and robotics Correll et al. (2016). SNNs leverage sparse spike features to replace the dense features used in ANNs, resulting in significant improvements in energy efficiency. However, the 3D point clouds exploration of SNNs remains quite limited, with existing approaches Lan et al. (2023b); Ren et al. (2023); Wu et al. (2024b) demonstrating inadequate performance. Furthermore, the architecture has yet to be designed to effectively integrate the biological characteristics and the spatial information. It is intriguing to explore the application of Spiking Transformers in the processing of point clouds.

However, using villina SSA and Spikformer Zhou et al. (2023c) to process 3D features is not ideal. There are primarily three challenges. Firstly, salient points representations located at object edges or at boundaries between different categories are critical for accurate prediction. However, SSA tends to focus on highly similar points, smoothing even neglecting salient features, consequently degrading performance. Secondly, unlike the smaller number of tokens in 2D visual tasks, point clouds usually consist of a large number of points, making the use of SSA to model global dependence computationally prohibitive. Furthermore, considering the redundancy in point cloud data, SSA is incapable of simultaneously capturing both local and global features. Overall, 3D features call for a redesigned, efficient attention mechanism that highlights discriminative (edge) features and seamlessly scales from local to global point-cloud feature modeling.

To address the above challenges, we propose a Spiking Discrepancy Attention Mechanism (SDAM), which uses the spiking differences between the Query and Key as the attention matrix to enhance the representation capability for complex spatial structures. It consists of Spiking Element Discrepancy Attention (SEDA) and Spiking Intensity Discrepancy Attention (SIDA), applied in the shallow and deep stages, respectively. The SEDA captures local feature relationships between central points and neighboring points through fine-grained element-wise spiking differences. In contrast, the SIDA models global dependencies using coarse-grained differences in spiking intensity among central points. Additionally, we design a Spatially-Aware Spiking Neuron that effectively encodes spatial information in the initial membrane potential, compensating for the loss of spatial information in spike representations. Ultimately, we integrate SDAM with Spatially-Aware Spiking Neuron into a hierarchical multi-stage local-global architecture, termed as Spiking Discrepancy Transformer, which is shown in Figure 1. The main contributions are as follows:

- We propose a Spiking Discrepancy Attention Mechanism (SDAM) tailored to the characteristics of point clouds. The mechanism includes Spiking Element Discrepancy Attention (SEDA) and Spiking Intensity Discrepancy Attention (SIDA), which effectively represent complex local-global spatial information.
- We design a Spatially-Aware Spiking Neuron that encodes spatial information in the initial membrane potential, thereby compensating for the loss of spatial information in spike representations.
- The Spiking Discrepancy Transformer achieves state-of-the-art performance among SNN-based approaches. Besides, our method's theoretical energy consumption is significantly lower compared to ANN-based approaches.

## 2   RELATED WORK

**Deep SNNs and Spiking Transformers.** Numerous studies focus on learning methods and architectures for deep SNNs. Spatio-temporal backpropagation (STBP) Wu et al. (2018) enables the direct training of SNNs via backpropagation across both spatial and temporal dimensions. Temporal backpropagation Kheradpisheh & Masquelier (2020) calculates the gradients of spike timings with respect to the membrane potential at the moment of spike generation. Threshold-dependent batch normalization (tdBN) Zheng et al. (2021) facilitates the training of deeper SNNs. Additionally, SEW-ResNet Fang et al. (2021b) proposes a spiking element-wise residual mechanism to enhance the performance of deep SNNs. Spikformer Zhou et al. (2023c) is the first to transform all components of the Vision Transformer into a spike-based formulation, thereby pioneering the integration of SNNs with Transformer. Spike-driven Transformer Yao et al. (2024b) introduces a linear-complexity spike-driven self-attention for efficiency. The following works focus on structural enhancements Zhou et al. (2023a;b); Wang et al. (2023a); Zhou et al. (2024a;b), training methodologies Wang et al. (2023b), and applications across different tasks Bal & Sengupta (2024). Research on the application of Spiking Transformers across various fields is gradually unfolding.

**Point Cloud Analysis.** Point-based methods are pioneering in point cloud processing tasks, with numerous studies exploring 3D spatial convolutions Li et al. (2018); Yan et al. (2020), point cloud spatial encoding Mohammadi & Salarpour (2024), and network architecture design Ma et al. (2022). Recently, the application of Transformers in the point cloud have demonstrated significant potential, with the Point Transformer Guo et al. (2021); Zhao et al. (2021) being a notable milestone. To enhance computational efficiency, PatchFormer Zhang et al. (2022) proposes Patch Attention, Point Transformer V2 Wu et al. (2022) introduces Grouped Vector Attention, and Flatformer Liu et al. (2023) further proposes Flattened Window Attention. The Point Transformer V3 Wu et al. (2024c) achieves state-of-the-art performance across various tasks. In SNNs, Spiking PointNet Ren et al. (2023) incorporates spiking neurons into the PointNet framework, but its performance remains suboptimal. P2SResLNet Wu et al. (2024b) combines 3D kernel point convolutions with spiking neurons, and E-3DSNN Qiu et al. (2024a) introduces a Spike Sparse Convolution to efficiently extract sparse 3D features. However, these methods diverge from the biological simplicity and dynamics inherent in traditional SNNs. The ANN-to-SNN conversion approach Lan et al. (2023a) also yields subpar results. Spiking Point Transformer Wu et al. (2024a) relies on single local feature extraction and fails to capture the global characteristics, leading to performance degradation. Currently, there is an urgent need for a coherent design from spiking neurons to spiking attention to enhance the capabilities of the 3D Spiking Transformer.

## 3   METHOD

### 3.1   PRELIMINARIES

**Spiking Neuron** is the basic unit of SNNs. We select the typical hard-reset Leaky-Integrate-and-Fire (LIF) neuron Wu et al. (2018) as example,

$$H[t] = (1 - \frac{1}{\tau})V[t-1] + \frac{1}{\tau}X[t], \tag{1}$$

$$S[t] = \Theta(H[t] - V_{\text{th}}), \tag{2}$$

$$V[t] = H[t](1 - S[t]) + V_{\text{reset}}S[t], \tag{3}$$

where $\tau$ is the membrane time constant, and $X[t]$ is the input current received at the time step $t$. When the membrane potential $H[t]$ surpasses the threshold $V_{\text{th}}$, the spiking neuron will trigger a spike $S[t]$ to subsequent layers. The heaviside step function $\Theta(v)$ is defined as 1 for $v \geq 0$ and 0 for $v < 0$. $V[t]$ represents the post-spike membrane potential, which is either $H[t]$ if no spike is generated or is reset to $V_{\text{reset}}$ upon a spike event. It is worth noting that $V[0]$ is typically considered to be zero, whereas different insights may apply in 3D tasks.

**Spiking Self-Attention** is different from vanilla self-attention Vaswani et al. (2017). Given the spike-form input $\mathbf{X}$, the Query $\mathbf{Q}$, Key $\mathbf{K}$, and Value $\mathbf{V}$ are in spike form. Besides, it discards the softmax normalization for the attention map, which can be described by the following equation:

$$\mathbf{Q} = \mathcal{SN}_{\mathbf{Q}}(\text{L-BN}_{\mathbf{Q}}(\mathbf{X})), \mathbf{K} = \mathcal{SN}_{\mathbf{K}}(\text{L-BN}_{\mathbf{K}}(\mathbf{X})), \mathbf{V} = \mathcal{SN}_{\mathbf{V}}(\text{L-BN}_{\mathbf{V}}(\mathbf{X})), \tag{4}$$

$$\text{SSA}(\mathbf{Q}, \mathbf{K}, \mathbf{V}) = \mathcal{SN}(\mathbf{Q}\mathbf{K}^{\text{T}}\mathbf{V} * s), \tag{5}$$

where $\mathcal{SN}$ denotes the spiking neuron layer and L-BN represents that the features pass through Linear and Batch Normalization sequentially. $s$ is the scaling factor.

## 3.2 SPIKING DISCREPANCY ATTENTION MECHANISM

The spatial features of unordered, irregular point clouds fundamentally differ from the semantic features of ordered, structured 2D visual data. SSA guides feature aggregation through dot product similarity, allowing points with similar features to be assigned higher weights. However, in 3D scenes, discriminative edge regions often exhibit stark changes in local geometric features, resulting in lower feature similarity scores. The smooth aggregation of SSA can lead to a dilution of scores in these critical regions, subsequently causing edge blurring effects. While spiking-based modeling brings efficiency, it inevitably introduces information loss. The presence of channel-wise spiking dot product mismatches within SSA exacerbates this issue, further degrading its representation capability. Moreover, the dot product-based SSA does not inherently satisfy translation invariance, which is crucial for 3D tasks. In summary, 3D analysis requires an efficient attention mechanism that focuses on distinct features and easily adapts from local to global point-cloud modeling. We design a Spiking Discrepancy Attention Mechanism (SDAM) tailored for 3D analysis,

$$\text{SDAM}(\mathbf{Q}, \mathbf{K}, \mathbf{V}) = \text{SD}(\mathbf{Q}, \mathbf{K}) \circ \mathbf{V}, \tag{6}$$

where SD is defined as the spiking discrepancy, which is a spike-driven feature metric and can be obtained through subtraction between spike sequences. $\circ$ represents a spike-driven matrix operator which varies depending on the scope of the attention modeling. SD simulates the cortical neurons' response mechanism to asynchronous multi-channel spike misalignment through spike discrepancy sensitivity, where lateral inhibition of neighboring neuron activity enhances edge contrast. It also satisfies the translation invariance of spiking features, as shown in Table 1 by its ability to robustly recognize point clouds after spatial transformations.

Inspired by the multi-stage hierarchical architecture, we extend SDAM into a hierarchical combination of Spiking Element Discrepancy Attention (SEDA) and Spiking Intensity Discrepancy Attention (SIDA). Specifically, SEDA captures the fine-grained spiking variation trends between the center point and its neighboring points within a local point cloud cluster, while SIDA depicts the coarse-grained significant differences at a macro scale among point cloud. The details will be elaborated in the following subsections.

Table 1: Comparison of model performance degradation after spatial transformations on the datasets. The types of spatial transformations include random rotation, translation, and scaling. PT refers to Point Transformer. OA and $\text{OA}_\text{T}$ respectively represent the original accuracy and the accuracy on the transformed datasets.

| Models | Modelnet40 | | ScanObjectNN | |
|---|---|---|---|---|
| | OA(%) | OA$_\text{T}$(%) | OA(%) | OA$_\text{T}$(%) |
| PT Zhao et al. (2021)(ANN) | 93.7 | 90.7(-3.0) | 86.4 | 83.2(-3.2) |
| PTV3 Wu et al. (2024c)(ANN) | 94.5 | 92.3(-2.2) | 87.9 | 85.8(-2.1) |
| SSA Zhou et al. (2023c)(SNN) | 89.8 | 86.2(-3.6) | 83.2 | 80.4(-2.8) |
| **SDAM**(SNN) | 92.5 | 91.2(-1.3) | 86.2 | 85.0(-1.2) |

## 3.3 SPIKING ELEMENT DISCREPANCY ATTENTION

As shown in Figure 1, the Spiking Element Discrepancy Attention (SEDA) explicitly addresses the limitation of geometric feature dilution in edge regions by leveraging channel-wise spiking difference sensitivity, a bio-inspired mechanism that mimics lateral inhibition in cortical neurons Mao & Massaquoi (2007).

Unlike conventional dot-product similarity that prioritizes smooth feature aggregation, SEDA operates on a fundamental hypothesis: local geometric discriminability arises from spiking misalignment between neighboring points. The explanation is provided in the Appendix C. Formally, given a central query point spiking feature $\mathbf{q} \in \mathbb{S}^{T \times C}$, its $n$ neighboring key points spiking feature set $\mathbf{k} = \{\mathbf{k}_j \in \mathbb{S}^{T \times C} \mid j = 1, 2, \cdots, n\}$ and central value point spiking feature $\mathbf{v}$, where $T$ represents time steps and $C$ denotes the number of channels, SEDA computes pairwise multi-channel spiking discrepancy,

$$\mathbf{SD}_j = \mathbf{q} - \mathbf{k_j}, \tag{7}$$

$$\mathbf{SD}(\mathbf{q}, \mathbf{k}) = \mathcal{SN}\left(\sum_{j=1}^{n} \mathbf{SD}_j * s\right), \tag{8}$$

$$\mathbf{SEDA}(\mathbf{q}, \mathbf{k}, \mathbf{v}) = \mathbf{SD}(\mathbf{q}, \mathbf{k}) \odot \mathbf{v}, \tag{9}$$

where SD is denoted as spiking difference and $\mathcal{SN}$ represents the spiking neuron. $s$ is a synaptic scaling factor. $\odot$ is the element-wise masking. $*$ means element-wise multiplication.

To intuitively demonstrate the spatial representation capability of SEDA, we perform t-SNE on the feature matrix obtained after applying SEDA and SSA on the ShapeNetPart Yi et al. (2016).

As shown in Figure 2, SEDA demonstrates superior geometric discriminability compared to SSA. Specifically, SEDA induces two critical clustering properties: 1) Intra-part feature compactness, where points belonging to the same object part (e.g., chair legs or surfaces) form tightly cohesive clusters, and 2) Inter-part margin amplification, exhibiting enlarged separation distances between clusters corresponding to distinct geometric components. In contrast, SSA features exhibit diffused distributions with overlapping clusters across object parts, indicating geometric ambiguity. These results indicate that the network incorporating SEDA successfully learns detailed spatial position features.

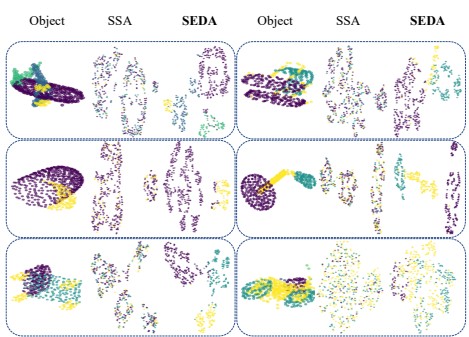

Figure 2: t-SNE visualization of SSA and SEDA. Points of different colors correspond to distinct components of the object. We compress the features produced by SSA and SEDA in the first stage of the network.

### 3.4 SPIKING INTENSITY DISCREPANCY ATTENTION

Building upon SEDA's local geometric discriminability, we propose Spiking Intensity Discrepancy Attention (SIDA) to capture global structural patterns through spiking intensity divergence, as shown in Figure 1. Spiking intensity represents the cumulative value of spikes for the current position features in a purely additive form. Unlike SEDA that focuses on micro-scale spiking discrepancy between neighbors, SIDA operates on a macro-scale spatial hypothesis: topological significance emerges from population-level firing intensity contrast across different point cloud regions. The explanation is provided in the Appendix C. Formally, given the $N$-points global spiking feature $\mathbf{Q}, \mathbf{K}, \mathbf{V} \in \mathbb{S}^{T \times N \times C}$, SIDA can be written as follows:

$$\mathbf{SD}(\mathbf{Q}, \mathbf{K}) = \mathcal{SN}((\sum^{C} \mathbf{Q} - \sum^{C} \mathbf{K}^{T}) * s), \tag{10}$$

$$\mathbf{SIDA}(\mathbf{Q}, \mathbf{K}, \mathbf{V}) = \mathbf{SD}(\mathbf{Q}, \mathbf{K}) \cdot \mathbf{V}. \tag{11}$$

By capturing global spiking intensity differences, SIDA effectively identifies macro-structural patterns in point clouds, particularly the overall geometric layout and key object components. Specifically, SIDA's sensitivity to intensity differences addresses two critical challenges in spiking-based point cloud analysis:

1) The intrinsic information loss in spiking features may compromise accurate 3D perception, while spiking intensity offers a statistically robust solution for modeling 3D salient disparity regions. 2) The inherent translation invariance of intensity divergence aligns seamlessly with 3D geometric priors, enhancing the model's ability to understand spatial structures. To demonstrate that SIDA captures global features more effectively than SSA, we visualized the spiking point features, as shown in the Figure 3. SIDA produces sparse spike activations at critical locations of the point cloud skeleton, such as the wings, tail, and nose of an aircraft; the lampshade, lamp post, and lamp base of a lamp. In contrast, SSA focuses on less important regions or repeatedly emphasizes a particular component, such as the fuselage of an aircraft or only the lampshade of a lamp.

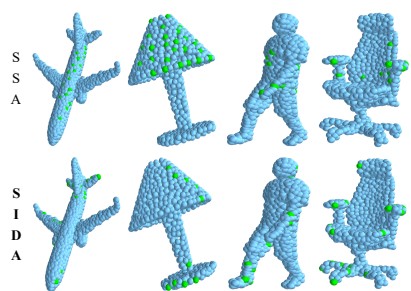

Figure 3: Visualization of SSA and SIDA feature map. The green point indicates each spike, while blue denotes silence. The feature map is derived from the spike matrix obtained after applying SSA or SIDA in the final stage.

The hierarchical synergy between SEDA and SIDA forms a bio-plausible computational mechanism for point cloud analysis. SEDA enhances local geometric discriminability in early stages, while SIDA models global topological structures through spiking intensity contrast in deep stages. This dual-attention framework enables the network to capture both fine-grained local details and global geometric relationships, improving object structure understanding.

## 3.5 SPATIALLY-AWARE SPIKING NEURON

Spiking features suffer from information loss when modeling complex spatial positions, and this issue becomes more severe as the depth of the network increases. To this end, we propose a Spatially-Aware Spiking Neuron (SASN) based on membrane potential dynamics, inspired by the characteristics of neuronal membrane potential dynamics and the specific traits of 3D tasks. The Initial Membrane Potential (IMP) affects neuronal dynamics Shen et al. (2024b). Therefore, embedding spatial information into the IMP can enhance spatio-temporal perception.

Specifically, due to the decoupling of spatial point sampling from the neural network, we can encode the selected point coordinates into the IMP of spiking neurons using trigonometric functions Zhang et al. (2023b) in a non-learnable manner before the network processing stage. In the first time step, according to Eq. 1, the membrane potential changes from $\frac{1}{\tau}X[1]$ to:

$$H[1] = (1 - \frac{1}{\tau})P[0] + \frac{1}{\tau}X[1], \tag{12}$$

where $P[0]$ represents the value of the position encoding corresponding to the specific spiking neuron. Details on IMP position encoding can be found in the Appendix A. SASN leverages the injection of spatial information from the IMP to achieve the spatio-temporal information interaction in SNN-based point cloud processing. SASN can directly replace conventional spiking neurons, and we also conduct an ablation study on the insertion position.

## 3.6 OVERALL ARCHITECTURE

Based on the SEDA, SIDA and SASN, we construct a hierarchical Spiking Discrepancy Transformer (SDT), which is shown in Figure 1. SDT is an encoder structure consisting of multiple stages that progressively downsample and model the point set. Each stage contains the Spiking Sub-sampling Unit (SSU) and SDT block. SSU performs point set sampling and embedding transformation. Point set sampling involves downsampling and center-nearest point sampling, implemented using Farthest Point Sampling (FPS) and K-Nearest Neighbors (KNN), respectively. FPS and KNN are common sampling methods without training parameters. For stage $l$, the output of point set sampling includes the central point features $\mathbf{X}_l \in \mathbb{R}^{T \times N \times C}$, corresponding neighboring point features $\mathbf{X}_l^k \in \mathbb{R}^{T \times N \times C \times k}$, and the original coordinates $\mathbf{P}_l \in \mathbb{R}^{N \times 3}$. The embedding transformation is formulated as:

$$\mathbf{X}_l' = \text{Linear}(\mathcal{SN}(\text{CAT}(\mathbf{X}_l, \mathbf{X}_l^k))), \tag{13}$$

$$\mathbf{Y}_l = \text{MP}(\mathbf{X}_l') + \text{AP}(\mathbf{X}_l'), \tag{14}$$

where CAT represents aggregation. After aggregating, the spiking points are projected into the neuron membrane potential $\mathbf{X}_l'$. Finally, by using Max-Average Pooling (MP, AP), we aggregate the local features from the neighborhood onto $\mathbf{Y}_l$. Each SDT block contains SDAM and Multi-layer Perceptron (MLP). First half of the stages employ the SEDA to extract local point cloud details and the others utilize the SIDA to extract global point cloud information. For 3D classification, the number of stages is set to 4. SDT by default employs the membrane potential shortcut residual Hu et al. (2023). Additional architectural details are presented in Appendix B.

# 4 EXPERIMENTS

## 4.1 EXPERIMENTAL SETTINGS

Building upon prior research in SNNs for point cloud analysis, we initially evaluate the performance of the proposed SDT for 3D classification. This evaluation encompasses experiments on both the synthetic ModelNet40 Wu et al. (2015) and the real-world ScanObjectNN Uy et al. (2019). Furthermore, we showcase the versatility of SDT by extending its application to 3D semantic segmentation and object part segmentation tasks. For semantic segmentation, we employ the Stanford Large-Scale 3D Indoor Spaces (S3DIS) Armeni et al. (2016) which present significant challenges. For object part segmentation, we utilize the ShapeNetPart Yi et al. (2016). To comprehensively assess

the performance of SDT, we compare it with both conventional ANN methods and cutting-edge SNN approaches, including direct training methodologies and ANN-to-SNN conversion strategies. Finally, we perform an ablation study to systematically analyze the contributions of individual components within SDT, thereby validating the efficacy of our proposed framework.

Table 2: Performance and theoretical energy consumption on 3D classification. ∗ means self implementation

| Method | Type | Param(M) | ModelNet40 | | | ScanObjectNN | | |
|---|---|---|---|---|---|---|---|---|
| | | | OA(%)↑ | mAcc(%)↑ | Energy(mJ)↓ | OA(%)↑ | mAcc(%)↑ | Energy(mJ)↓ |
| PointNet Qi et al. (2017a) [CVPR17] | ANN | 3.47 | 89.20 | 86.00 | 2.07 | 68.20 | 63.40 | 2.07 |
| PointNet++ Qi et al. (2017b) [NeurIPS17] | ANN | 1.74 | 91.90 | 89.10 | 18.72 | 77.90 | 75.40 | 18.71 |
| KPConv Thomas et al. (2019) [ICCV19] | ANN | 15.20 | 92.10 | 90.70 | 94.53 | 85.30 | 83.69 | 94.50 |
| PointTransformer Zhao et al. (2021) [ICCV21] | ANN | 9.58 | 93.70 | 90.60 | 84.64 | 86.01 | 84.10 | 84.07 |
| PointMLP Ma et al. (2022) [ICLR22] | ANN | 12.60 | 94.10 | 91.30 | 72.38 | 85.40 | 83.90 | 72.36 |
| Point-GPT Chen et al. (2024) [NeurIPS23] | ANN | 19.46 | 94.00 | 91.03 | 20.48 | 86.90 | 85.17 | 20.47 |
| PointGT Zhang et al. (2024c) [TMM24] | ANN | - | 92.60 | 90.00 | - | 86.50 | 84.90 | - |
| PointNet-SNN Lan et al. (2023b) [ICCV23] | ANN-to-SNN | 3.50 | 88.17 | 84.02 | 0.26 | 66.56 | 60.33 | 0.27 |
| KPConv-SNN Wu et al. (2024b) [AAAI24] | ANN-to-SNN | 15.20 | 70.50 | 67.60 | - | 43.90 | 38.70 | - |
| Spiking PointNet Ren et al. (2023) [NeurIPS23] | SNN | 3.50 | 88.61 | 84.20 | 0.24∗ | 65.40 | 61.30 | 0.28∗ |
| P2SResLNet Wu et al. (2024b) [AAAI24] | SNN | 15.20 | 90.60 | 89.20 | - | 81.20 | 79.40 | - |
| E-3DSNN Qiu et al. (2024a) [AAAI25] | SNN | 3.27 | 91.70 | 88.40 | 1.76∗ | 83.91∗ | 81.92∗ | 2.64∗ |
| SPT Wu et al. (2024a) [AAAI25] | SNN | 9.64 | 91.43 | 89.39 | 13.3 | 82.23 | 80.12 | 13.5∗ |
| **SDT (T=1)** | SNN | 2.25 | 92.18 | 88.92 | 0.45 | 85.25 | 83.20 | 0.61 |
| **SDT (T=4)** | SNN | 2.25 | 92.46 | 89.48 | 1.33 | 86.19 | 84.37 | 2.11 |

**Implementation Details.** We implement SDT using PyTorch Paszke et al. (2019) on four RTX 4090 GPUs. For the neuron models, we utilize those provided by the SpikingJelly library Fang et al. (2023). We construct the code framework based on Zhang et al. (2023a). Additionally, we specify the hyper-parameters used in our experiments, as summarized in Table 3. They are based on the common practices in the ANNs with slight adjustments. The initial point number refers to the quantity of point clouds fed into the net-

Table 3: Hyper-parameters of SDT on various datasets.

| Parameter | ModelNet40 | ScanObjectNN | S3DIS | ShapeNetPart |
|---|---|---|---|---|
| Learning Rate | $5e-2$ | $1.5e-3$ | $1e-2$ | $2e-3$ |
| Weight Decay | $1e-4$ | $5e-2$ | $1e-4$ | $1e-4$ |
| Batch Size | 32 | 32 | 96 | 48 |
| Training Epochs | 300 | 300 | 100 | 200 |
| Optimizer | SGD | AdamW | AdamW | Adam |
| Initial Point Number | 1024 | 1024 | 24000 | 2048 |

work during the training process. In addition, the theoretical energy consumption formulation is provided in the Appendix D.

## 4.2 3D CLASSIFICATION

**Data and Metric.** The ModelNet40 Wu et al. (2015) dataset contains 12,311 CAD models with 40 object categories. They are split into 9,843 models for training and 2,468 for testing. We follow the data preparation procedure of Qi et al. Qi et al. (2017b) and uniformly sample the points from each CAD model. While ModelNet40 is widely regarded as the standard benchmark for point cloud analysis, its synthetic nature and the rapid advancement in point cloud methods may limit its relevance for modern approaches. Therefore, we also evaluate our method on the ScanObjectNN benchmark Uy et al. (2019). ScanObjectNN includes approximately 15,000 objects, categorized into 15 classes with 2,902 unique real-world object instances. This dataset presents significant challenges for point cloud analysis due to factors such as background interference, noise, and occlusions. In our experiments, we focus on the most challenging perturbed variant called PB_T50_RS. For evaluation metrics, we use the mean Accuracy (mAcc) within each category and the Overall Accuracy (OA) over all classes. The training and inference speed are provided in the Appendix E.1.

**Comparison with SNNs.** SDT significantly outperforms existing SNN approaches in Table 2. For instance, the previous best model, E-3DSNN Qiu et al. (2024a), achieves 91.70% OA on ModelNet40 with 3.27M parameters, and SPT Wu et al. (2024a) reaches 82.23% OA on ScanObjectNN. In contrast, with only 2.25M parameters, SDT surpasses these models by 0.76% on ModelNet40 and 3.96% on ScanObjectNN, achieving both enhanced computational efficiency and improved performance.

**Comparison with ANNs.** As shown in Table 2, when compared to ANN-based methods such as, PointMLP Ma et al. (2022), Point-GPT Chen et al. (2024), and PointGT Zhang et al. (2024c), which utilize floating-point representations to encode richer information, SDT delivers competitive performance with only a slight decrease in accuracy. Furthermore, our focus is on leveraging SDT to minimize energy consumption during 3D point cloud processing while maintaining competitive classification accuracy. SDT achieves overall accuracies of 92.46% and 86.19% on two benchmark

datasets, respectively, closely matching the performance of PointMLP (94.10%) and Point-GPT (86.90%). Remarkably, SDT consumes only 1.8% of the energy consumption of PointMLP (1.33 mJ VS. 72.38 mJ) on ModelNet40, 10.3% of the Point-GPT (2.11 mJ VS. 20.47 mJ) on ScanObjectNN. SDT also outperforms ANN-based models such as PointNet++ and KPConv, requiring fewer parameters and less energy, demonstrating potential of SNNs for efficient 3D point cloud processing.

## 4.3 SEMANTIC AND OBJECT PART SEGMENTATION

**Data and Metric.** For the semantic segmentation task, we conduct experiments on S3DIS Armeni et al. (2016). The S3DIS dataset, designed for semantic scene parsing, consists of 271 rooms spanning six areas across three buildings. In accordance with prior works Tchapmi et al. (2017); Qi et al. (2017b); Zhao et al. (2021), area 5 is excluded from training and reserved for testing. Following the standard evaluation protocol Qi et al. (2017b), we employ mIoU, mean class-wise accuracy (mAcc), and overall point-wise accuracy (OA) as evaluation metrics. For the object part segmentation task, we use the ShapeNetPart Yi et al. (2016). It consists of 16,880 models from 16 shape categories, with 14,006 3D samples for training and 2,874 for testing. The number of parts for each category is between 2 and 6, with 50 different parts in total. We use the sampled point sets produced by Qi et al. Qi et al. (2017b) for a fair comparison with prior work. For evaluation metrics, we report category mIoU and instance mIoU.

Table 4: Semantic segmentation results on S3DIS, evaluated on Area 5.

| Method | Type | OA(%) | mAcc(%) | mIoU(%) | ceiling | floor | wall | beam | column | window | door | table | chair | sofa | bookcase | board | clutter | Param(M) | Energy(mJ) |
|---|---|---|---|---|---|---|---|---|---|---|---|---|---|---|---|---|---|---|---|
| PointNet Qi et al. (2017a) | ANN | – | 49.0 | 41.1 | 88.8 | 97.3 | 69.8 | 0.1 | 3.9 | 46.3 | 10.8 | 59.0 | 52.6 | 5.9 | 40.3 | 26.4 | 33.2 | 3.5 | 5.5 |
| TangentConv Tatarchenko et al. (2018) | ANN | – | 62.2 | 52.6 | 90.5 | 97.7 | 74.0 | 0.0 | 20.7 | 39.0 | 31.3 | 77.5 | 69.4 | 57.3 | 38.5 | 48.8 | 39.8 | 1.5 | 32.3 |
| PointCNN Li et al. (2018) | ANN | 85.9 | 63.9 | 57.3 | 92.3 | 98.2 | 79.4 | 0.0 | 17.6 | 22.8 | 62.1 | 74.4 | 80.6 | 31.7 | 66.7 | 62.1 | 56.7 | 46.2 | 324.5 |
| PCCN Wang et al. (2018) | ANN | – | 67.0 | 58.3 | 92.3 | 96.2 | 75.9 | 0.3 | 6.0 | 69.5 | 63.5 | 66.9 | 65.6 | 47.3 | 68.9 | 59.1 | 46.2 | – | – |
| PAT Yang et al. (2019) | ANN | – | 70.8 | 60.1 | 93.0 | 98.5 | 72.3 | 1.0 | 41.5 | 85.1 | 38.2 | 57.7 | 83.6 | 48.1 | 67.0 | 61.3 | 33.6 | 9.3 | 97.2 |
| PointWeb Zhao et al. (2019) | ANN | 87.0 | 66.6 | 60.3 | 92.0 | 98.5 | 79.4 | 0.0 | 21.1 | 59.7 | 34.8 | 76.3 | 88.3 | 46.9 | 69.3 | 64.9 | 52.5 | 4.8 | 68.1 |
| HPEIN Jiang et al. (2019) | ANN | 87.2 | 68.3 | 61.9 | 91.5 | 98.2 | 81.4 | 0.0 | 23.3 | 65.3 | 40.0 | 75.5 | 87.7 | 58.5 | 67.8 | 65.6 | 49.4 | – | – |
| MinkowskiNet Choy et al. (2019) | ANN | – | 71.7 | 65.4 | 91.8 | 98.7 | 86.2 | 0.0 | 34.1 | 48.9 | 62.4 | 81.6 | 89.8 | 47.2 | 74.9 | 74.4 | 58.6 | – | – |
| KPConv Thomas et al. (2019) | ANN | – | 72.8 | 67.1 | 92.8 | 97.3 | 82.4 | 0.0 | 23.9 | 58.0 | 69.0 | 81.5 | 91.0 | 75.4 | 75.3 | 66.7 | 58.9 | 20.4 | 136.5 |
| PointTransformer Zhao et al. (2021) | ANN | 90.8 | 76.5 | 70.4 | 94.0 | 98.5 | 86.3 | 0.0 | 38.0 | 63.4 | 74.3 | 89.1 | 82.4 | 74.3 | 80.2 | 76.0 | 59.3 | 4.9 | 76.8 |
| PTv2 Wu et al. (2022) | ANN | 91.1 | 77.9 | 71.6 | – | – | – | – | – | – | – | – | – | – | – | – | – | 12.8 | 400.1 |
| PTv3 Wu et al. (2024c) | ANN | 91.7 | 79.0 | 73.6 | 92.4 | 98.3 | 86.6 | 0.0 | 55.8 | 63.7 | 77.1 | 83.8 | 93.3 | 79.1 | 79.4 | 85.4 | 61.7 | 46.2 | 687.7 |
| E-3DSNN Qiu et al. (2024a) | SNN | 89.8 | 73.3 | 67.4 | 95.3 | 98.5 | 82.3 | 0.0 | 28.0 | 55.8 | 71.5 | 81.2 | 89.8 | 69.2 | 76.4 | 67.0 | 61.6 | 10.9 | 14.4 |
| SDT | SNN | 90.1 | 76.8 | 69.6 | 93.8 | 98.5 | 84.8 | 0.0 | 42.1 | 57.0 | 69.7 | 77.6 | 91.3 | 76.1 | 74.1 | 79.3 | 59.9 | 10.7 | 7.3 |

**Performance Comparison.** The results are presented in Table 4 and Table 5. We compare our work with the previous state-of-the-art ANN domain. Since no SNN has yet reported results on the S3DIS and ShapeNetPart datasets, we test the performance of the SNN state-of-the-art method E-3DSNN Qiu et al. (2024a). For effectiveness and fairness, we use the case where $T$=1. On the S3DIS, our model achieves a mIoU of 69.6%, while E-3DSNN arrives at only 67.4%. On the ShapeNetPart, our model achieves a 2.0% improvement in category mIoU and a 1.3% improvement in instance mIoU compared to E-3DSNN, obtaining the SOTA in the SNN area. Furthermore, our method has competitive performance, approaching the SOTA accuracy of 73.6% mIoU on S3DIS and 86.6% instance mIoU on ShapeNetPart achieved by ANN-based methods, while the energy consumption is only 1.06% (7.29mJ vs 687.68mJ) and 3.73% (4.74mJ vs 126.96mJ) of ANN-SOTAs. We provide the training curve on S3DIS in the Appendix E.3.

Table 5: Object Part Segmentation results on ShapeNetPart.

| Method | Type | cat. mIoU(%) | ins. mIoU | Param(M) | Energy(mJ) |
|---|---|---|---|---|---|
| PointNet Qi et al. (2017a) | ANN | 80.4 | 83.7 | 8.3 | 26.5 |
| PCCN Wang et al. (2018) | ANN | 81.8 | 85.1 | - | - |
| PointNet++ Qi et al. (2017b) | ANN | 81.9 | 85.1 | 1.7 | 22.5 |
| DGCNN Wang et al. (2019) | ANN | 82.3 | 85.1 | 1.5 | 23.1 |
| SpiderCNN Xu et al. (2018) | ANN | 81.7 | 85.3 | 2.2 | 41.4 |
| PointConv Wu et al. (2019) | ANN | 82.8 | 85.7 | 1.7 | 15.4 |
| PointCNN Li et al. (2018) | ANN | 84.6 | 86.1 | 46.4 | 328.7 |
| KPConv Thomas et al. (2019) | ANN | 85.1 | 86.4 | 20.7 | 144.5 |
| PointTransformer Zhao et al. (2021) | ANN | 83.7 | 86.6 | 7.8 | 127.0 |
| PointMLP Ma et al. (2022) | ANN | 84.6 | 86.1 | 5.2 | 54.2 |
| PointGPT Chen et al. (2024) | ANN | 84.1 | 86.2 | 6.8 | 102.3 |
| E-3DSNN Qiu et al. (2024a) | SNN | 81.7 | 83.8 | 4.9 | 8.8 |
| SDT | SNN | 83.7 | 85.1 | 4.6 | 4.7 |

## 4.4 ABLATION STUDY

**Ablation on Spatially-Aware Spiking Neuron.** As shown in Figure 4, we investigate the effectiveness of integrating SASN into different components (two types of attention (SEDA and SIDA) and the downsampling block SSU). The baseline architecture uses Spiking Self-Attention (SSA) and the common LIF neurons. The results demonstrate that SASN enhances the model's spatial perception capabilities, particularly proving more effective in refining point cloud representations through SSU, SEDA, and SIDA. Besides, we compare other neurons with SASN in the Appendix E.5.

**Ablation on Attentions and Hierarchical Framework.** As shown in Table 6, rows 1-4 demonstrate that both SIDA and SEDA are more effective than vanilla SSA in modeling point clouds with nearly identical parameters. Rows 5 further indicate that, compared to using SIDA or SEDA individually, a multi-level architecture where SEDA is employed in the early stages to refine local point cloud

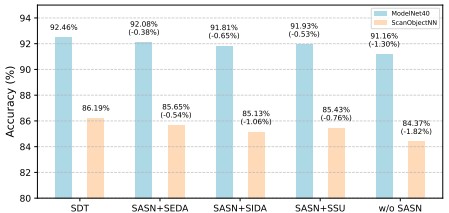

Figure 4: The ablation study for SASN. We observe that the incorporation of SASN into various modules consistently results in enhanced performance.

Table 6: Ablation study on attention variants in different stages. The numbers marked in green and orange represent the use of SEDA and SIDA, respectively, in the corresponding stage. SSA denotes the framework where all stages employ the SSA, while "None" refers to remove attention mechanism.

| Attention Type | Stage Index (1-4) | ModelNet40 | ScanObjectNN |
|---|---|---|---|
| None | 1,2,3,4 | 85.25% | 78.04% |
| SSA | 1,2,3,4 | 89.78% | 83.20% |
| SEDA | 1,2,3,4 | 92.34% | 85.45 % |
| SIDA | 1,2,3,4 | 92.13% | 84.73% |
| SEDA+SIDA | 1,2,3,4 | **92.46%** | **86.19%** |

representations and SIDA is utilized in the later stages to extract global information—achieves superior performance in point cloud processing. More results are provided in the Appendix E.2.

Table 7: Time Step Ablation.

| Time Step | ModelNet40 | | ScanObjectNN | |
|---|---|---|---|---|
| | OA(%) | mAcc(%) | OA(%) | mAcc(%) |
| 1 | 92.18 | 88.92 | 85.25 | 83.20 |
| 2 | 92.34 | 89.31 | 83.87 | 81.82 |
| 4 | **92.46** | **89.48** | **86.19** | **84.37** |
| 6 | 91.93 | 88.71 | 85.53 | 83.70 |

Table 8: Ablation study on network depth and width.

| Depth | Width | Param(M) | ModelNet40 | | ScanObjectNN | |
|---|---|---|---|---|---|---|
| | | | OA(%)↑ | Energy(mJ)↓ | OA(%)↑ | Energy(mJ)↓ |
| 2 | [24, 48] | 0.20 | 89.78 | 0.25 | 82.99 | 0.66 |
| 2 | [48, 96] | 0.32 | 90.21 | 0.35 | 83.96 | 0.79 |
| 2 | [96, 192] | 0.73 | 90.71 | 0.53 | 84.47 | 1.03 |
| 4 | [24, 48, 96, 192] | 0.72 | 91.87 | 0.89 | 85.43 | 1.66 |
| 4 | [48, 96, 192, 384] | 2.25 | 92.46 | 1.33 | 86.19 | 2.11 |
| 4 | [96, 192, 384, 768] | 8.07 | 92.69 | 4.56 | 86.45 | 7.85 |
| 6 | [24, 48, 96, 192, 384, 768] | 9.95 | 92.04 | 5.87 | 86.20 | 9.07 |
| 6 | [48, 96, 192, 384, 768, 1536] | 30.52 | 92.42 | 17.73 | 86.08 | 20.29 |

**Time Step.** As depicted in Table 7, increasing time steps $T$ within a certain range improves accuracy. We set the maximum $T$ to 6 in our study, the results are peaking at $T=4$ with 92.46% on the ModelNet40 and 86.19% on the ScanObjectNN. As discussed in Wu et al. (2024b), the results can be attributed to the scarcity of temporal cues in 3D datasets. Thus, increasing $T$ might lead to redundant computations without enhancing informative representation. Small time steps may be suitable for practical applications.

**Ablation on Network Depth and Width.** Table 8 compares the parameter count, performance, and energy consumption of the SDT under varying depths and widths. The results demonstrate that excessively increasing either width or depth in point cloud tasks does not yield significant performance gains but escalates both model size and computational cost. Balancing the accuracy and efficiency, we adapt a configuration with 4-stage depth and [48, 96, 192, 384] for width.

**Sparsity Analysis.** As demonstrated in Table 9, we analyze the sparsity of each component in the SIDA and SEDA, and their sparsity further corroborates the low energy consumption of SDT.

**Robustness Analysis.** In the Appendix E.4 and E.6, we further analysis the robustness of SDT.

## 4.5 VISUALIZATION RESULTS

Figure 5 presents the Semantic Segmentation of SDT on the S3DIS dataset. The predictions exhibit a high degree of similarity to the ground truth, highlighting the effectiveness of our architecture in segmentation tasks. More visualization results are shown in the Appendix F.

Table 9: The Sparsity of each Attention Type. **Q**, **K**, **V** means Query, Key, Value Matrices, **A** means Attention Map, **O** means the results of Self-Attention.

| Attention Type | **Q**(%) | **K**(%) | **V**(%) | **A**(%) | **O**(%) |
|---|---|---|---|---|---|
| SSA | 3.12 | 5.54 | 9.46 | 6.82 | 5.90 |
| SEDA | 1.06 | 1.24 | 1.15 | 3.82 | 4.69 |
| SIDA | 1.55 | 1.48 | 1.56 | 3.75 | 4.93 |

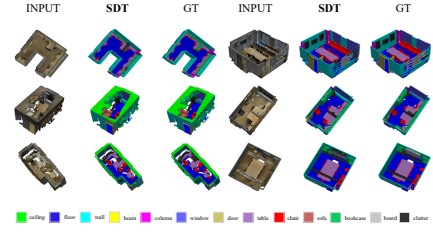

Figure 5: Visualization of segmentation on S3DIS. GT means the ground truth.

## 5 CONCLUSION

For 3D tasks, we design a Spiking Discrepancy Attention Mechanism (SDAM), which includes Spiking Element Discrepancy Attention and Spiking Intensity Discrepancy Attention to model local-global spatial features. A Spatially-Aware Spiking Neuron is designed to align with the SDAM. Based on these, we propose the hierarchical Spiking Discrepancy Transformer (SDT). SDT achieves SOTA performance within SNNs and exhibits theoretically lower energy consumption compared to ANNs. SDT can further solidify the foundation for exploring the application of SNNs in 3D tasks, and also promote the design of next-generation neuromorphic chips for point cloud processing. The limitations and future work are discussed in the Appendix G

## 6 ACKNOWLEDGMENTS AND DISCLOSURE OF FUNDING

We sincerely thank Dr. Wei Fang for his assistance with this work. This work is supported by grants from the National Natural Science Foundation of China (62236009, 62206141, 62027804, and 62425101), and the major key project of the Pengcheng Laboratory (PCL2021A13). Computing support was provided by Pengcheng Cloudbrain.

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

## A    DETAILS OF INITIAL MEMBRANE POTENTIAL

We choose the x-coordinate $x_j$ of the $j$-th point $p_j = (x_j, y_j, z_j) \in \mathbb{R}^3, j \in [1, N]$ in the $N$ points as an example.

$$\mathbf{U}^x[j] = \text{IMP}(x_j), \mathbf{U}^x[j] \in R^{\frac{2C}{3}}, \tag{15}$$

$$\mathbf{U}^x[j][m] = \begin{cases} \sin\left(x_j \cdot \alpha/(t \cdot \beta^{\frac{3m}{2C}})\right), m = 2n \\ \cos\left(x_j \cdot \alpha/(t \cdot \beta^{\frac{3m}{2C}})\right), m = 2n+1 \end{cases} \tag{16}$$

$$\mathbf{U}[j] = \text{Concat}(\mathbf{U}^x[j], \mathbf{U}^y[j], \mathbf{U}^z[j]), \mathbf{U}[j] \in R^C. \tag{17}$$

where $C$ means the channels, $m \in [0, \frac{C}{3}]$, $\alpha$ and $\beta$ control the amplitude and wavelength of the trigonometric functions, respectively, and are set to 1000 and 100 in experiments. $t$ is the current time step, used to distinguish between different time points in the temporal dimension. Integrating the three coordinates, $p_j$ is encoded into $\mathbf{U}[j]$. The dimensional expanses from 3 to $C$. This parameter-free positional encoding maps the point cloud locations into high-frequency feature information, serving as the Initial Membrane Potential (IMP) for Spiking Neuron, which mitigates the SNN's inherent information loss.

## B    ARCHITECTURE DETAILS

### B.1    POINT CLOUD CLASSIFICATION

Our Hierarchical Spiking Discrepancy Transformer (SDT) framework for point cloud classification is shown in Fig. 1 in the main text. Herein, we provide additional details regarding the model. We use four stages. Each SSU module reduces the cardinality of the point set to $\frac{1}{2}$ in each stage. The K-Nearset Neighbor is set to 40. The initial encoding channels are set to 48, and the expansion in each stage is [2, 2, 2, 1]. For classification, we also perform a global MAP (Max-Average Pooling) over the pointwise features to get a global feature for the whole point set. This global feature is passed through a Spiking MLP with a linear layer to get the global classification logits. We apply SEDA in the first half stages and SIDA in the last half stages.

### B.2    POINT CLOUD SEGMENTATION

For dense prediction tasks such as semantic segmentation, we adopt a U-Net in which the encoder described in the main text is coupled with a Spiking Feature Propagation decoder Qian et al. (2022). Consecutive stages in the decoder are connected by Spiking Points Propagation Unit. Their primary function is to map features from the downsampled input point set $P_2$ onto its superset $P_1 \supset P_2$. To this end, each input point feature is processed by a Spiking Linear layer, and then the features are mapped onto the higher-resolution point set $P_1$ via trilinear interpolation. These interpolated features from the preceding decoder stage are summarized with the features from the corresponding encoder stage, provided via a membrane shortcut skip connection.

For the segmentation head, the final decoder stage produces a feature vector for each point in the input point set. We also apply a Spiking MLP and a Linear layer to map this feature to the final logits. Besides, we use four stage for S3DIS Armeni et al. (2016) and ShapeNetpart Yi et al. (2016). In the S3DIS dataset, each SSU module reduces the cardinality of the point set to $\frac{1}{4}$ in each stage. The K-Nearset Neighbor is set to 32. The initial encoding channels are set to 48, and the expansion in each stage is [2, 2, 2, 1]. Each stage contains [4, 7, 4, 4] Transformer blocks, respectively. We apply SIDA in the final stage and SEDA in the other stages. In ShapeNetpart dataset, each SSU module reduces the cardinality of the point set to $\frac{1}{4}$ in the first stage while $\frac{1}{2}$ in the other stages. The K-Nearset Neighbor is set to 32. The initial encoding channels are set to 48, and the expansion in each stage is [2, 2, 2, 1]. Each stage contains only one Transformer block. We apply SEDA in the first half stages and SIDA in the last half stages.

## C    THEORETICAL ANALYSIS OF HYPOTHESES IN SEC 3.2

We provide a theoretical analysis of hypotheses in Sec 3.2. Prior ANN works have demonstrated that topological complexity of point clouds can be quantified by entropy Young & Wasserman (2001);

Tang et al. (2023); Jiang et al. (2021). Second, in the work Guo et al. (2022a), it has been shown that the information in spiking features can likewise be measured by entropy. In ANNs, the variations in the coordinates of point clouds cause the extracted features within the point-cloud network to differ; similarly, in SNNs the spiking features exhibit the same situation. It shows that the geometric features are related to spiking features. Thus, we further explain the hypotheses from the perspective of local and global entropy.

For SEDA, we propse hypothesis: "local geometric discriminability arises from spiking misalignment between neighboring points" at line 157. The validity of the SEDA hypothesis can be supported by analyzing the entropy and information content of local geometric features based on spiking misalignment within neighborhoods. Let $q \in \mathbb{S}^{T \times C}$ be the spiking feature of a query point, and $\{\mathbf{k}_j\}_{j=1}^n$ be the spiking features of its neighboring points. We define the multi-channel spiking difference as $\mathrm{SD}_j = q - \mathbf{k}_j$, and measure its magnitude $d_j = \mathrm{SD}_j$. Normalizing over the neighborhood yields a probability distribution:

$$p_j = \frac{d_j}{\sum_{k=1}^n d_k}, \tag{18}$$

The local geometric entropy is then defined as:

$$H_{\mathrm{local}} = -\sum_{j=1}^n p_j \log p_j, \tag{19}$$

which quantifies the uncertainty of spiking misalignment distribution. A high entropy (nearly uniform $p_j$) indicates low geometric distinctiveness, while a low entropy (dominated by few large $d_j$) indicates salient local geometric features. Thus, the local geometric discriminability is inversely proportional to $H_{\mathrm{local}}$ and SEDA's purpose is to highlight the local geometric saliency.

For SIDA, we propse hypothesis: "topological significance emerges from population-level firing intensity contrast across different point cloud regions" at line 187. We formalize macro-scale hypothesis by introducing Global Center Entropy which is computed across multiple cluster center points. In SIDA, the spike intensity of a given central point $m$ is equivalent to the spike firing rate $p_m$. We can normalize the firing rates of all points and subsequently compute the Global-Center-Entropy $H_{\mathrm{global}}$ of point cloud $M$,

$$q_m = \frac{\rho_m}{\sum_k \rho_k}, \tag{20}$$

$$H_{\mathrm{global}} = -\sum_{m=1}^M q_m \log q_m, \tag{21}$$

When the entropy $H_{\mathrm{global}}$ is smaller, it indicates that the feature information content of the point cloud is higher. This suggests that the network representation can capture the macro-level differences in the point cloud features. For SIDA's operation $g_{mn}$, $\mathbf{SD}(m, n) = \mathrm{SN}(\rho_m - \rho_n) \propto \frac{|q_m - q_n|}{\max_k q_k - \min_k q_k}$. $\frac{1}{H_{\mathrm{global}}} \propto \sum_{m<n} \mathbf{SD}(m, n)$. The total weight in SIDA is inversely proportional to $H_{\mathrm{global}}$. SIDA assigns higher weights to centers with "large firing rate differences leading to lower entropy," thereby directly capturing the macro-level topological salience of the point cloud.

## D  THEORETICAL ENERGY CONSUMPTION

According to the general convention of SNNs Panda et al. (2020); Yao et al. (2023), we posit that the MAC and AC operations are executed on 45nm hardware Horowitz (2014), with energy consumption values of $EC_{\mathrm{MAC}} = 4.6\mathrm{pJ}$ and $EC_{\mathrm{AC}} = 0.9\mathrm{pJ}$ per operation, respectively. The theoretical Energy Consumption (EC) of ANNs can be derived as follows:

$$EC_{\mathrm{ANN}} = 4.6\mathrm{pJ} \times \mathrm{MACs}. \tag{22}$$

In SNNs, the AC operations can be obtained by multiplying the MAC operations by the firing rate $f$ of input spikes and the simulation time step $T$,

$$\mathrm{ACs} = \mathrm{MACs} \times f \times T. \tag{23}$$

In SpikingPoint, the operations of the first layer are MACs to map the floating-point positions of the point cloud to spike features, while subsequent-layers operations are ACs for modeling sparse spiking-point features,

$$EC_{\text{SP}} = 4.6\text{pJ} \times \text{MACs}^1 + 0.9\text{pJ} \times \sum_{l=2}^{L} \text{ACs}^l, \tag{24}$$

where $L$ denotes the number of linear layers in the SDT. Note that we ignore the energy of BN, as it can be incorporated into the linear layers during inference. Energy consumption for point cloud pre-processing is not accounted for, as it does not involve SNNs computations.

# E  MORE EXPERIMENTAL RESULTS

## E.1  TRAINING AND INFERENCE SPEED

Table 10: Performance and Speed comparison on the ScanObjectNN dataset. OA denotes Overall Accuracy.

| Model | Type | ScanObjectNN (OA(%)) | Throughput (ins./sec.) |
|---|---|---|---|
| PointNet | ANN | 68.20 | 4212 |
| PointNet++ | ANN | 77.90 | 1872 |
| KPConv | ANN | 85.30 | 1281 |
| PointTransformer | ANN | 86.01 | 188 |
| PointMLP | ANN | 85.40 | 191 |
| Point-GPT | ANN | 86.90 | 134 |
| PointGT | ANN | 86.50 | – |
| PointNext | ANN | 87.70 | 2040 |
| PointNet-SNN | ANN-to-SNN | 66.56 | 4188 |
| KPConv-SNN | ANN-to-SNN | 43.90 | 1267 |
| Spiking PointNet | SNN | 65.40 | 1391 |
| P2SResLNet | SNN | 81.20 | – |
| E-3DSNN | SNN | 83.91 | 245 |
| SPT | SNN | 82.23 | 168 |
| **SDT (ours)** | **SNN** | **86.19** | **279** |

As shown in Table 10, we analyze the runtime performance of various models on ScanObjectNN, along with their corresponding training and inference speeds. Our model, SDT, achieves state-of-the-art performance among SNNs while maintaining a competitive operational speed. When performing inference on GPUs, SNNs do not exhibit a clear throughput advantage over ANNs. However, SNNs are typically deployed on neuromorphic hardware, which significantly accelerates their execution and makes inference faster compared to ANNs.

Table 11: More ablation study on various spiking attentions implemented on different stages. The numbers marked in green and orange represent the use of SEDA and SIDA, respectively, in the corresponding stage. SSA denotes the framework where all stages employ the SSA, while "None" refers to the framework where all stages consist solely of MLPs without any attention mechanism.

| Attention Type | Stage Index (1-4) | ModelNet40 | ScanObjectNN |
|---|---|---|---|
| None | 1,2,3,4 | 85.25% | 78.04% |
| SSA | 1,2,3,4 | 89.78% | 83.20% |
| SEDA | 1,2,3,4 | 92.34% | 85.45 % |
| SIDA | 1,2,3,4 | 92.13% | 84.73% |
| SEDA+SIDA | 1,2,3,4 | **92.46%** | **86.19%** |
| SIDA+SEDA | 1,2,3,4 | 92.09% | 85.05% |
| SEDA+SIDA | 1,2,3,4 | 92.22% | 85.34% |
| SEDA+SIDA | 1,2,3,4 | 92.34% | 85.22% |
| SIDA+SEDA | 1,2,3,4 | 91.81% | 84.94% |
| SIDA+SEDA | 1,2,3,4 | 91.41% | 84.49% |

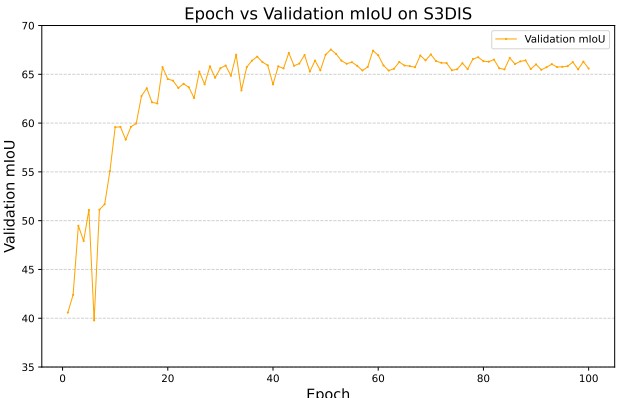

Figure 6: Visualization of training curve on the S3DIS dataset. GT means the ground truth. We choose the validation miou in each epoch as y-axis.

### E.2    ABLATION ON ATTENTIONS AND HIERARCHICAL FRAMEWORK

As shown in Table 11, rows 7,8 demonstrate that the SEDA and SIDA both play crucial roles. Excessive reliance on SEDA for extracting local information or on SIDA for extracting global information can lead to performance degradation. Rows 6, 9-10 shows that SEDA and SIDA cannot be inverted; extracting global information first and then focusing on local details is counterintuitive, and experiments have also demonstrated that this approach leads to a significant performance degradation. This further validates the effectiveness of SDT.

### E.3    EPOCH-ACCURACY CURVE

To further illustrate the performance of SDT, we present the validation mIoU curve on the S3DIS dataset, demonstrating the variation with increasing epochs, as shown in Figure 6. Tab. 4 in the main text reports the test mIoU results.

### E.4    CROSS-DATASET RESULTS

Table 12: Cross-dataset Results on Point Cloud Classification

| Architecture | Train: ModelNet40 Test: ScanObjectNN | Train: ScanObjectNN Test: ModelNet40 |
|---|---|---|
| PointNet Qi et al. (2017a) | 31.1 | 50.9 |
| SpiderCNN Xu et al. (2018) | 30.9 | 46.6 |
| PointNet++ Qi et al. (2017b) | 32.0 | 47.4 |
| DGCNN Wang et al. (2019) | 36.8 | 54.7 |
| PointCNN Li et al. (2018) | 24.6 | 49.2 |
| SimpleView Goyal et al. (2021) | 40.5 | 57.9 |
| **SDT** | **51.3** | **66.2** |

According to the experiments in work Goyal et al. (2021), we design cross-dataset evaluations based on our SDT. Our results surpass those of prior SNN-based point-cloud methods. This demonstrate that our SDT network exhibits strong cross-dataset generalization.

### E.5    ABLATION ON NEURON TYPES

As shown in Table 13, we conduct experiments under the setting T=4. Our SASN outperforms the neurons you highlighted on both ModelNet40 and ScanObjectNN, indicating that the spatial information encoded in SASN is better suited to point-cloud tasks.

Table 13: Ablation on various Neuron

| Neuron Type | ModelNet40 OA(%)/mAcc(%) | ScanObjectNN OA(%)/mAcc(%) |
|---|---|---|
| SASN | 92.46/89.48 | 86.19/84.37 |
| IMP Shen et al. (2024a) | 91.69/88.34 | 84.56/82.44 |
| RealSpike Guo et al. (2022b) | 91.88/88.69 | 85.01/83.45 |
| TernarySpike Guo et al. (2024) | 91.45/88.21 | 84.91/83.22 |
| MultiSpike Qiu et al. (2024a) | 91.96/89.01 | 85.34/83.81 |

## E.6 ROBUSTNESS ABLATION STUDY

Table 14: Robustnss Performance Comparison of SDT, SEDA, SIDA, SSA and ANN-based models

| Models | Type | $ACC_{clean}$ | $ACC_{noise}$ | Uniform | Gaus. | Impulse | Upsamp. | Bg. |
|---|---|---|---|---|---|---|---|---|
| PointNet Qi et al. (2017a) | ANN | 90.7 | 67.3 | 87.6 | 85.6 | 70.9 | 86.0 | 6.4 |
| PointNet++ Qi et al. (2017b) | ANN | 93.0 | 78.5 | 79.6 | 83.6 | 64.9 | 82.8 | 81.4 |
| DGCNN Wang et al. (2019) | ANN | 92.6 | 74.3 | 85.4 | 83.4 | 75.1 | 80.9 | 46.9 |
| PointMLP Ma et al. (2022) | ANN | 93.5 | 63.1 | 77.2 | 67.4 | 59.8 | 61.3 | 49.7 |
| PCT Guo et al. (2021) | ANN | 92.9 | 71.9 | 87.9 | 86.1 | 60.9 | 82.6 | 42.1 |
| Point Transformer Zhao et al. (2021) | ANN | 93.7 | 78.0 | 89.9 | 88.2 | 69.9 | 74.3 | 67.7 |
| PTV3 Wu et al. (2024c) | ANN | 94.5 | 86.0 | 91.3 | 90.0 | 77.4 | 86.8 | 84.4 |
| SSA Zhou et al. (2023c) | SNN | 89.8 | 86.9 | 88.8 | 89.0 | 86.4 | 84.8 | 85.7 |
| SEDA | SNN | 92.3 | 90.0 | 91.0 | 91.0 | 91.6 | 87.4 | 88.7 |
| SIDA | SNN | 92.1 | 87.8 | 90.4 | 90.0 | 89.3 | 84.2 | 85.3 |
| SDT | SNN | 92.5 | 90.4 | 91.2 | 91.5 | 91.8 | 88.2 | 89.2 |

Here, we analyze the noise robustness of the Spiking Discrepancy Attention Mechanism and its corresponding two attention variants, SEDA and SIDA. We compare them with SSA, as well as with the ANN-based models, especially Point-Transformer and Point-Transformer v3, as shown in Table 14. Following the ModelNet40-C Goyal et al. (2021) benchmark and previous work Zhang et al. (2024a), our model is trained on the clean ModelNet40 dataset Wu et al. (2015) and evaluated using its corrupted test sets. ModelNet40-C is constructed by applying various corruptions to the ModelNet40 test set, encompassing 15 distinct corruption types categorized into Density, Noise, and Transformation, each with five variations. Furthermore, each corruption type includes five severity levels. For our assessment, we focused on the noise category, specifically selecting Uniform, Gaussian, Impulse, Upsampling, and Background corruptions. We also use the overall accuracy for performance evaluation. The results demonstrate that SNNs combined with the Transformer architecture exhibit superior noise robustness. Within the category of SNNs, our designed SEDA and SIDA demonstrate enhanced noise robustness compared to the original SSA.

Table 15: Ablation studies and hyperparameter analysis on ModelNet40 and ScanObjectNN.

| Analysis Type | Item | Value | ModelNet40 OA(%)/mAcc(%) | ScanObjectNN OA(%)/mAcc(%) |
|---|---|---|---|---|
| **Ours (Default)** | | | **92.46/89.48** | **86.19/84.37** |
| Spike Generation | Poisson Encoding | - | 92.23/89.18 | 85.89/84.15 |
| Neuron Hyperparameters | Decay Factor | 0.25 | 92.43/89.51 | 86.22/84.21 |
| | | 1.0 | 92.75/89.77 | 85.89/84.41 |
| | Threshold | 0.5 | 92.56/89.91 | 85.91/84.02 |
| | | 1.5 | 92.26/89.11 | 86.01/84.17 |
| | Surrogate Function | Sigmoid | 92.13/88.94 | 85.85/83.69 |
| | | SoftSign | 91.99/88.41 | 85.56/83.91 |
| Spike Perturbations | Random Drop | - | 91.87/89.01 | 85.89/84.31 |
| | Random Flip | - | 91.56/88.39 | 85.23/83.45 |
| | Random Noise | - | 91.72/88.52 | 85.71/83.75 |

Besides, we conduct more comprehensive robustness evaluations under various scenarios in Table 15: altering the spike encoding—i.e., spike generation settings (Poisson Encoding), the spiking-neuron hyperparameters (Decay Factor, Threshold and Surrogate Function), and noise scenarios (perturbations and transformations in the spike feature). The "Spike Perturbations" refer to perturbations applied to the spiking features within the attention mechanism. "Random Drop" denotes the random dropping

of spike features during inference. "Random Flip" denotes the random flipping of spike features during inference. "Random Noise" denotes the addition of random noise to the spike features during inference. For hyperparameters, SDT adopts rate coding for spike generation, a neuron threshold of 1.0, a decay factor of 0.5, and uses Atan as the surrogate function. The results demonstrate that the SDT design is relatively robust to various hyperparameters and noisy scenarios, exhibiting strong generalization.

# F   MORE VISUALIZATION RESULTS

## F.1   VISUALIZATION OF SEDA

In Figure 7, we show more t-SNE visualization of spiking features, further proving that SEDA demonstrates superior geometric discriminability compared to the SSA.

## F.2   VISUALIZATION OF SIDA

In Figure 8, we show more spiking point features of SIDA and SSA to show that SIDA captures global features more effectively than SSA.

## F.3   VISUALIZATION OF S3DIS RESULTS

In Figure 9, we show more Semantic Segmentation results on the S3DIS dataset.

## F.4   VISUALIZATION OF SDAM AND SSA ATTENTION MAPS

In Figure 10, we compare the attention map distributions of SSA versus SDAM (SEDA and SIDA). The visualization results demonstrate that SDAM is capable of better capturing global point cloud information, particularly regarding the geometric distinguishability of edge features. This further validates the effectiveness of SDAM compared to SSA in point cloud tasks.

# G   LIMITATIONS AND FUTURE WORKS

As mentioned in Section 3.2, our SDAM and SDT are specifically designed for point cloud analysis. In the domains of images and text, these mechanisms may require further refinement to enhance their generalization capabilities. Future works will focus on designing SNNs for point cloud analysis in larger, open-world scenarios.

# H   USE OF LLMS

We declare that the LLMs are used solely to aid or polish the writing and are not involved in the development of the main methodology or comparative experiments.

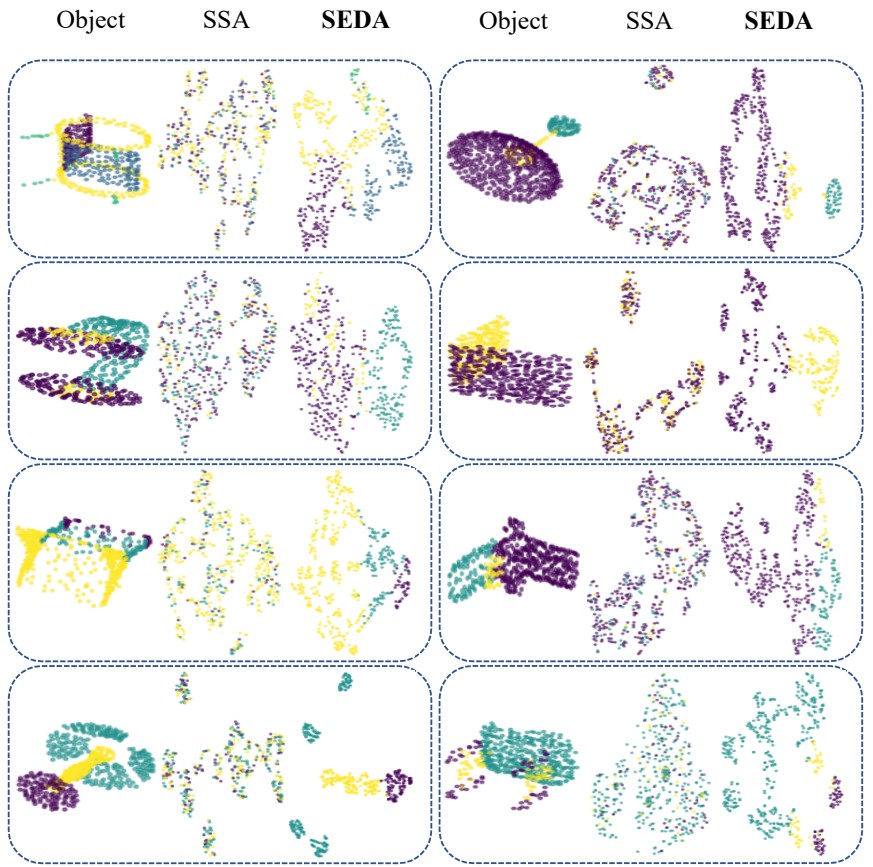

Figure 7: t-SNE visualization of SSA and SEDA. We utilize the feature maps obtained after the first stage of SSA or SEDA in the network and subsequently compress them into a two-dimensional plane using t-SNE.

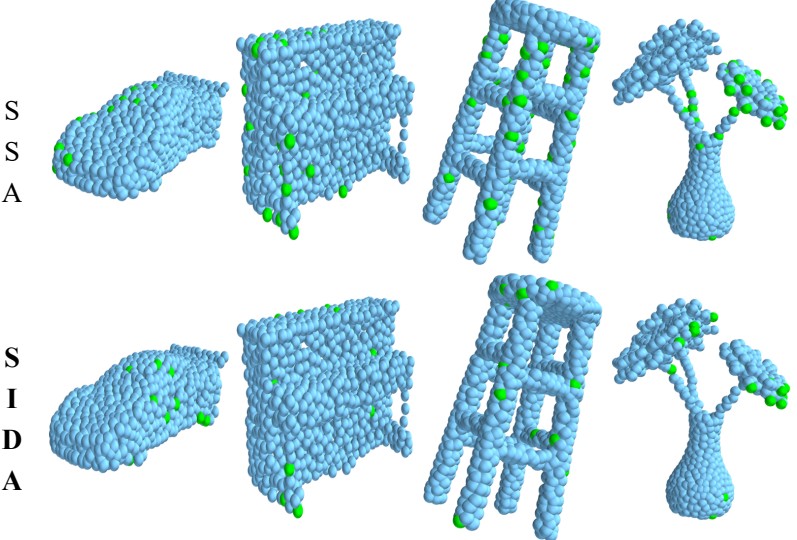

Figure 8: Visualization of SSA and SIDA feature map. The green point indicates each spike, while blue denotes silence. The feature map is derived from the spike matrix obtained after applying SSA or SIDA in the final stage.

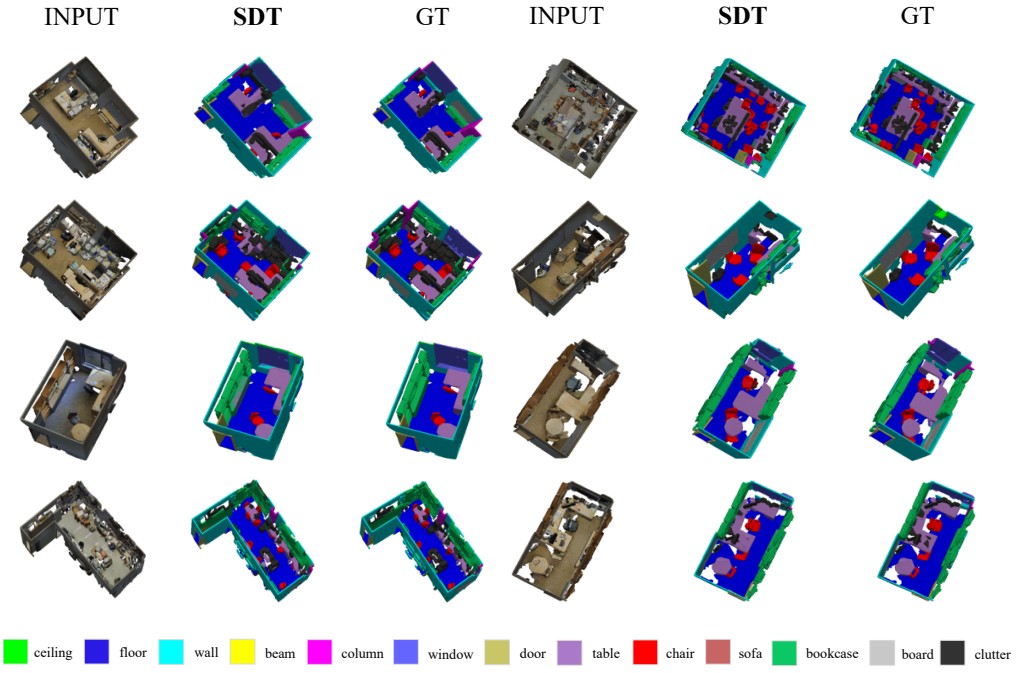

Figure 9: Visualization of semantic segmentation results on the S3DIS dataset. GT means the ground truth. We selected six representative scenarios for validation.

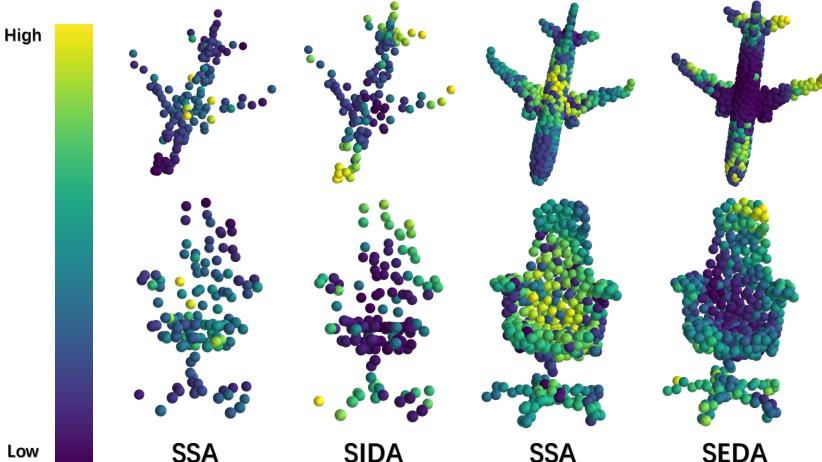

Figure 10: Visualization of SSA and SDAM Attention Map. High values indicate elevated attention levels, while low values suggest reduced focus. We observe that SSA tends to prioritize regions with high similarity, specifically the denser areas of the point cloud. However, these regions often fail to yield effective features for classification. In contrast, SDAM (comprising SEDA and SIDA) gravitates towards regions with significant disparity—specifically areas exhibiting geometric discriminability. Features derived from these regions prove to be highly effective for identifying point cloud categories. This demonstrates that the difference-based measurement employed in SDAM is more effective for point cloud analysis than the similarity-based measurement used in SSA.

