# OpenReview forum: "Spiking Discrepancy Transformer for Point Cloud Analysis"
_ICLR.cc/2026/Conference — ICLR 2026 Poster_

### Official Review · Reviewer_Aiaz · 2025-10-26

**Soundness:** 3
**Presentation:** 3
**Contribution:** 2
**Rating:** 6
**Confidence:** 4

**Summary:**

This paper proposes the Spiking Discrepancy Transformer (SDT) for 3D point cloud analysis, with key contributions including the Spiking Discrepancy Attention Mechanism (SDAM), which consists of local SEDA and global SIDA variants, the Spatially-Aware Spiking Neuron (SASN) that encodes spatial information through the initial membrane potential, and achieving strong performance in the SNN domain across multiple datasets while claiming significant energy reduction.

**Strengths:**

To address the challenges of existing SNNs in 3D point cloud tasks, a complete system is proposed, ranging from attention mechanisms to neuron design and overall architecture. Additionally, the experiments are comprehensive, covering multiple tasks including classification, semantic segmentation, and object part segmentation, with thorough ablation and visualization analysis.

**Weaknesses:**

1. The paper claims SSA applied to 3D point cloud tasks overlooks significant features and fails to capture local and global information, but lacks clear theoretical or experimental validation to support these claims.

2. The paper asserts that "spiking discrepancy" better captures geometric discriminability but does not justify why disparity is more suitable for point clouds than similarity, relying solely on biological plausibility without rigorous proof.

3. The presentations of SEDA and SIDA are not sufficiently clear: symbols are inconsistent, key variables and steps are ambiguously defined.

**Questions:**

1.A more rigorous theoretical analysis or appropriate references must be provided to substantiate the challenges associated with applying SSA to 3D point cloud tasks.

2.It is essential to validate the sensitivity of spiking discrepancy to edge features using synthetic data, or alternatively, to analyze and compare the attention map distributions of SSA versus SDAM in order to substantiate the fundamental claims made in the paper.

3.In Eq. 8, why are the neighbors not weighted? The simple summation approach may allow distant neighbors to exert an undue influence, which requires further justification.

4.The selection of the scaling factor s is not discussed, and the lack of ablation studies to support its choice undermines the robustness of the methodology.

---

> ### Author Response · Authors · 2025-11-20
> **Rebuttal-1**
>
> We appreciate your detailed comments. We would like to address your concerns below.
>
> ### Weaknesses and Questions:
> > ***Weakness 1, Question 1**: A more rigorous theoretical analysis or appropriate references must be provided to substantiate the challenges associated with applying SSA to 3D point cloud tasks. For example, the paper claims SSA applied to 3D point cloud tasks overlooks significant features and fails to capture local and global information.*
>
> **A1**: We appreciate your question. In the Introduction of our manuscript, we articulated three primary reasons regarding the inadequacy of Spiking Self-Attention (SSA) for point cloud analysis:
> 1. The representation of salient points—specifically those located at object edges or boundaries between distinct categories—is critical for accurate prediction. However, SSA tends to prioritize highly similar points, leading to the smoothing or even neglect of these salient features, which consequently degrades performance.
>
> We provided a rigorous theoretical analysis based on information entropy in Appendix C, elucidating why the "spike disparity" mechanism outperforms traditional "similarity-based" mechanisms in point cloud tasks:
> Local Level (SEDA): We define a local geometric entropy, $H_{local}$, and demonstrate that geometric discriminability is positively correlated with the non-uniformity (i.e., low entropy) of spike feature disparity. By emphasizing neighbors with significant disparities, SEDA effectively minimizes $H_{local}$, thereby enhancing the representation of edges and critical structural features.
> Global Level (SIDA): We define a global central entropy, $H_{global}$, and establish that topological saliency is positively correlated with the non-uniformity (i.e., low entropy) of firing intensities. SIDA reduces $H_{global}$ by amplifying intensity differences, thereby facilitating a focused attention on key components.
>
> The limitations of Spiking Self-Attention (SSA) in point cloud analysis fundamentally stem from its reliance on dot-product similarity, where the attention weights are formulated as:$$w_j \propto \langle q, k_j \rangle = \sum_{c} q_{c} \cdot k_{j,c}$$In geometric data, feature vectors $k_j$ at structural edges often differ significantly from the center point $q$, causing dot products to minimize and the normalized probabilities $p_j^{SSA}$ to approach zero; conversely, smooth regions exhibit high feature similarity, leading to a uniform weight distribution where $p_j^{SSA} \approx 1/n$. Consequently, SSA induces consistently high local entropy, defined as:$$H_{local}^{SSA} = - \sum_j p_j^{SSA} \log p_j^{SSA}$$Under this formulation, $H_{local}^{SSA}$ approaches $\log n$ in smooth areas and remains elevated at edges due to the suppression of geometrically significant but dissimilar neighbors, thereby inhibiting the expression of discriminative information—in sharp contrast to our proposed SEDA, which utilizes disparity ($p_j \propto \|q - k_j\|$) to minimize entropy.
>
> This deficiency extends to global modeling, where attention weights are predicated on aggregate similarity. In this regime, SSA disproportionately weights dense, homogeneous regions (e.g., fuselage) with high cumulative similarity over sparse, unique keypoints (e.g., wingtips), resulting in a diffuse activation distribution and elevated global entropy that effectively obscures topological saliency.
> Thus, dot-product similarity mechanisms tend to yield uniform attention distributions, which inadvertently increases these entropy values and obscures information regarding abrupt geometric transitions. Consequently, our approach relies not solely on biological plausibility but aligns fundamentally with the geometric information-theoretic nature of point clouds.
>
> 2. Unlike 2D visual tasks which involve a relatively small number of tokens, point clouds typically consist of a large number of points, rendering the use of SSA for modeling global dependencies computationally prohibitive. Specifically, the computational complexity of the original SSA is $2TN^2 C$. In contrast, our Spiking Enhancing-Difference Attention (SEDA) operates with a complexity of $TNKC$, where $K \ll N$. Furthermore, for our Spiking Interaction-Difference Attention (SIDA), the complexity is $\mathcal{O}(TN^2 + TN^2 C)$, which is approximately half that of SSA. Consequently, our proposed SDAM demonstrates higher computational efficiency.
>
> 3. Given the inherent redundancy in point cloud data, SSA is incapable of simultaneously capturing both local and global features effectively. First, based on the information entropy analysis presented in 1., SSA fails to effectively extract point cloud information at both local and global levels. Furthermore, SSA operates with uniform granularity, whereas SEDA and SIDA possess distinct global scale differentiation. This structural distinction further substantiates our argument.

---

> ### Author Response · Authors · 2025-11-20
> **Rebuttal-2**
>
> ### Weaknesses and Questions:
> > ***Weakness 2, Question 2**: The paper asserts that "spiking discrepancy" better captures geometric discriminability but does not justify why disparity is more suitable for point clouds than similarity. It is essential to validate the sensitivity of spiking discrepancy to edge features using synthetic data, or alternatively, to analyze and compare the attention map distributions of SSA versus SDAM in order to substantiate the fundamental claims made in the paper.*
>
> **A2**: In the first point of Response A1, we provide a theoretical analysis explaining why disparity is more suitable for point cloud processing than similarity. To further substantiate this, we include the attention map distributions for both SSA and SDAM in the revised manuscript. We will show them in Appendix F.4 in the manuscript.
>
>
> > ***Weakness 3**: The presentations of SEDA and SIDA are not sufficiently clear: symbols are inconsistent, key variables and steps are ambiguously defined.*
>
> **AW3**: Sorry for the confusion. In Eq. 8 and 10, * means element-wise multiplication. In Eq. 11, $\cdot$ means matrix multiplication. We will explain them in the manuscript.
>
> > ***Question 3**: In Eq. 8, why are the neighbors not weighted? The simple summation approach may allow distant neighbors to exert an undue influence, which requires further justification.*
>
> **AQ3**: In SEDA, the $Q$ and $K$ are already in the form of spikes.  Consequently, in Eq. 8, the value of each $SD_j$ is constrained to $-1$, $0$, or $1$.  We argue that the influence of inter-neighbor weights is already implicitly captured within these spike differences. The experimental results in Tab. 2 and the visualizations in Fig. 2 in the manuscript demonstrate that SEDA can model spike spatial differences. This validates the effectiveness of our design. In the context of SNNs for point cloud tasks, developing more flexible, spike-based inter-neighbor weighting mechanisms remains a direction for future work.
>
>
> > ***Question 4**: The selection of the scaling factor s is not discussed, and the lack of ablation studies to support its choice undermines the robustness of the methodology.*
>
> **A4**: Thanks for your advice. For SEDA, we select $s$=1.0 for all the stages. This is because the SEDA mechanism is fundamentally a linear attention method that does not involve matrix multiplication, thereby obviating the need for a small scaling factor to reduce variance. For SIDA, in emulation of ANN Transformers, we select $s$ as the number of attention heads divided by the number of feature channels for each stage. Furthermore, we conduct ablation studies to verify the role of $s$, with the results presented in Table R3-1. The experimental results indicate that for SEDA, the choice of $s$ has a negligible influence on the outcome. Conversely, for SIDA, a large $s$ leads to excessive variance and can still cause performance degradation, potentially due to issues such as gradient vanishing. This demonstrates the robustness of our method.
>
> #### Table R3-1
> | SEDA $s$ | SIDA $s$ |   ModelNet40 | ScanObjectNN |
> | -------- |:--------:|:--------:|:--------:|
> |  |  |OA(%)/mAcc(%)|OA(%)/mAcc(%)|
> | 1.0  | 1/16, 1/32, 1/64, 1/64 | 92.46/89.48    | 86.19/84.37    |
> | 0.5  | 1/16, 1/32, 1/64, 1/64 | 92.38/89.31    | 86.09/84.13    |
> | 0.125  | 1/16, 1/32, 1/64, 1/64 | 92.41/89.35  | 85.97/84.39    |
> | 1.0 | 0.125, 0.125, 0.125, 0.125| 92.18/88.91  | 85.84/84.09    |
> | 1.0 | 0.5, 0.5, 0.5, 0.5        | 91.97/88.76  | 85.62/83.89    |
> | 1.0 | 1.0, 1.0, 1.0, 1.0        | 91.83/88.51  | 85.44/83.65    |

---

### Official Review · Reviewer_krS2 · 2025-10-28

**Soundness:** 2
**Presentation:** 3
**Contribution:** 3
**Rating:** 6
**Confidence:** 5

**Summary:**

The paper introduces a Spiking Discrepancy Transformer (SDT) designed to enhance processing of 3D point clouds using spiking neural principles combined with self-attention. Traditional Spiking Transformers have mainly focused on 2D visual tasks, but 3D point clouds pose additional challenges due to their spatial disorder and scale. To address this, the authors propose a Spiking Discrepancy Attention Mechanism (SDAM), which measures differences in spike features to highlight key spatial information. Experiments show that the proposed method achieves state-of-the-art performance among SNNs, while significantly reducing energy consumption and parameter count.

**Strengths:**

1. Method innovation: The paper presents a hierarchical Spiking Discrepancy Transformer (SDT) that combines bio-inspired spiking dynamics with multi-scale attention to capture both local and global 3D features. Its spiking discrepancy and spatially-aware design enhance feature discriminability, efficiency, and spatial robustness for accurate 3D perception.
2. Paragraph Clarity: The paper demonstrates excellent narrative coherence and logical flow, presenting its ideas in a clear, progressive manner. The transitions between concepts are smooth, and the reasoning from problem to solution is both rigorous and intuitively convincing, enhancing the overall readability and scientific impact.
3. Experiment Solidity: The researchers conducted classification and segmentation tasks, enabling the model capture 3D feature robust and accurate.

**Weaknesses:**

1. Although multiple timesteps are assessed, the model still handles static point clouds, leaving the added value of temporal encoding somewhat uncertain.
2. The ablation experiment in Figure 4 does not explicitly indicate the baseline model and makes it difficult to validate the effectiveness of using SASN alone.
3. It would strengthen the paper to provide a clearer explanation of why SNNs are particularly advantageous for point cloud tasks, beyond potential energy efficiency, and to demonstrate this advantage empirically.

**Questions:**

1. The model operates on static point clouds, and the benefits of temporal encoding remain marginal.
2. More Results should be included in Figure 4 to prove the effectiveness of SASN.
3. Provide a clearer explanation of why SNNs are particularly advantageous for point cloud tasks.

---

> ### Author Response · Authors · 2025-11-20
> **Rebuttal**
>
> Thank you for your insightful feedback. Below, we provide our responses to the issues you raised.
>
> ### Weaknesses and Questions:
> > ***Weakness 1, Question 1**: Although multiple timesteps are assessed, the model still handles static point clouds, leaving the added value of temporal encoding somewhat uncertain.*
>
> **A1**: We appreciate the reviewer's inquiry. The temporal time step is an intrinsic characteristic of SNNs, and evaluating performance across multiple time steps is a standard convention in SNN research, including static image classification. While SDT demonstrates performance improvements as $T$ scales from 1 to 4, these gains are relatively modest. This behavior aligns with findings from prior SNN-based point cloud studies (e.g., SpikingPointNet, P2SResLNet, E-3DSNN, and SPT; refer to Table 2 in the manuscript). These observations further validate the suitability of SNNs for low-latency point cloud analysis.
>
>
>
> > ***Weakness 2, Question 2**: The ablation experiment in Figure 4 does not explicitly indicate the baseline model and makes it difficult to validate the effectiveness of using SASN alone.*
>
>
> **A2**: Sorry for the confusion caused by unclear experiments statement. We will explain the meaning of each compared model in Figure 4 and clarify our baseline model. In Figure 4, SDT denotes our final model. SASN+SEDA applies SASN only within the SEDA module. SASN+SIDA applies SASN only within the SIDA module. SASN+SSU applies SASN only within the SSU module. w/o SASN removes SASN in all modules.
>
> Our baseline removes both SEDA and SIDA, replaces them with the classic Spiking Self-Attention（SSA）, and uses the common LIF neurons. The corresponding results and the individual effectiveness of SASN are reported in Tab R2-1.
> #### Table R2-1
> | Architecture| ModelNet40 | ScanObjectNN |
> | -------- |:--------:|:--------:|
> |  |OA(%)/mAcc(%)|OA(%)/mAcc(%)|
> | Baseline（SSA）  | 88.20/86.23    | 81.97/79.86    |
> | Baseline + SASN  | 89.78/87.81     | 83.20/81.05    |
>
>
>
> > ***Weakness 3, Question 3**: It would strengthen the paper to provide a clearer explanation of why SNNs are particularly advantageous for point cloud tasks, beyond potential energy efficiency, and to demonstrate this advantage empirically.*
>
> **A3**: Thanks for your comments. Beyond energy efficiency, SNNs demonstrate stronger robustness than ANNs when handling point-cloud tasks. Since SNNs use binary 0-1 features, they are inherently resistant to 3D transformations and small perturbations in point-cloud scenes. As shown in manuscript's Tab. 1, Tab. 11 (Appendix D.3), and Tab. 15 (Appendix E.6), under transformations, various noise, or random noise, flip, drop injected into the attention module, SNNs, especially our proposed SDT exhibit smaller performance degradation. This characteristic is particularly crucial for real-world deployment.

---

> > ### Comment · Reviewer_krS2 · 2025-11-27
> >
> > Thanks. I believe most of my concerns are well-addressed. I consider rasing my score to 8.

---

### Official Review · Reviewer_UkDr · 2025-11-01

**Soundness:** 3
**Presentation:** 3
**Contribution:** 3
**Rating:** 6
**Confidence:** 4

**Summary:**

This paper proposes a hierarchical Spiking Discrepancy Transformer. Building upon spike feature discrepancies, authors construct a spiking discrepancy attention mechanism and design both spiking element discrepancy attention and spiking intensity discrepancy attention. Additionally, this work also introduce spatially-aware spiking neuron. Experimental results demonstrate the superior performance of the proposed model.

**Strengths:**

1. The paper is well-written and provides a detailed presentations of the methodology and motivations.
2. The proposed model demonstrates superior performance on both point cloud classification and segmentation tasks, also significantly reduce parameter count and energy consumption.
3. The proposed spiking discrepancy attention is well-motivated with good theoretical foundations, and its effectiveness has been validated through comprehensive experiments.

**Weaknesses:**

1. A core contribution of this work lies in the design of spiking discrepancy attention. This fundamental innovation requires further validation across diverse Transformer architectures to demonstrate its generalizability.

2. The use of KNN and trigonometric functions in the spatially-aware spiking neuron raises concerns that the proposed model architecture may not be truly spike-driven[1].

3. The semantic segmentation experiments lack comparisons with SNN models.

4. Table 6 shows limited performance gains when comparing SEDA+SIDA against SEDA-only. Therefore, I am concerned that the computational burden introduced by SIDA may not justify such marginal performance improvements.

[1] Spike-driven Transformer V2: Meta Spiking Neural Network Architecture Inspiring the Design of Next-generation Neuromorphic Chips. ICLR24

**Questions:**

Please refer to the Weaknesses.

---

> ### Author Response · Authors · 2025-11-20
> **Rebuttal**
>
> Thank you for your detailed comments and suggestions for improvement. We would like to address your concerns and answer your questions below.
>
> ### Weaknesses:
> > ***Weakness 1**: The Spiking Discrepancy Attention Mechanism (SDAM) requires further validation across diverse Transformer architectures to demonstrate its generalizability.*
>
> **A1**: As presented in Table 2 in the manuscript, we have already conducted comparisons with ANN-based Transformer architectures, such as PointTransformer, Point-GPT, and Point-GT as well as with SPT, an existing SNN-based Transformer framework method. Following your suggestion, we have further benchmarked our SDAM against attention mechanisms employed in various image-oriented Spiking Transformers. As shown in Table R1-1, we compared our method with Spike-Driven Self-Attention (from Spike-driven Transformer V2[1]), Dual Spike Self-Attention (from SpikingResformer[2]), and QK-Attention (from QKFormer[3]). The results clearly demonstrate the effectiveness of our proposed SDAM.
>
> #### Table R1-1
> | Model |ModelNet40 | ScanObjectNN |
> | -------- |:--------:|:--------:|
> |   |OA(%)/mAcc(%)|OA(%)/mAcc(%)|
> | SDAM  |92.46/89.48|86.19/84.37|
> | SPT |91.43/89.39|82.23/80.12|
> | SpikingResformer  |90.51/88.32|83.20/81.80|
> | Spike-Driven Transformer V2  |90.31/88.11|83.41/81.52|
> | QKFormer  |89.35/86.17|81.79/79.83|
>
>
> > ***Weakness 2**: The use of KNN and trigonometric functions in the spatially-aware spiking neuron raises concerns that the proposed model architecture may not be truly spike-driven[1].*
>
>
> **A2**: In the SASN architecture, point cloud coordinate downsampling and k-Nearest Neighbors (KNN) operations at each stage can be pre-processed prior to network inference. Consequently, the initial voltage $P[0]$ in Eq.  12 (derived in Appendix A), which involves trigonometric functions, can be pre-calculated. This eliminates the need for trigonometric computations during the inference phase. Regarding feature-level KNN operations, which rely on indexing based on coordinate-level KNN indices, we acknowledge that inference cannot be implemented solely using asynchronous neuromorphic hardware. However, KNN is a standard operation in point-based point cloud processing and can be efficiently executed via a dedicated hardware accelerator (e.g. [4]), with the outputs subsequently transmitted to the neuromorphic chip. Future work will focus on achieving the fully end-to-end deployment of SNNs for point cloud tasks entirely on neuromorphic hardware.
>
>
> > ***Weakness 3**: The semantic segmentation experiments lack comparisons with SNN models.*
>
> **A3**: First, we note that among existing SNN-based point cloud studies, only E-3DSNN provides experimental results for semantic segmentation; consequently, we included E-3DSNN’s results in Table 4 in the manuscript. Following your suggestion, we now provide the semantic segmentation performance of additional baselines in Table R1-2. These results demonstrate that our method achieves state-of-the-art (SOTA) performance.
>
> #### Table R1-2
> | Architecture| OA(%) | mAcc(%) | mIoU(%)|
> | -------- |:--------:|:--------:|:--------:|
> | SDT  | 90.1 | 76.8    |  69.6  |
> | E-3DSNN  | 89.8  |  73.3    |  67.4   |
> | SPT  | 89.4  |  73.1    |  66.6    |
> | P2SResLNet   | 88.4  |  71.8    |  64.6    |
>
>
> > ***Weakness 4**: Table 6 shows limited performance gains when comparing SEDA+SIDA against SEDA-only. Therefore, I am concerned that the computational burden introduced by SIDA may not justify such marginal performance improvements.*
>
> **A3**: Thank you for your question. The marginal gains in classification (Table 6) are attributed to performance saturation on small scale inputs (1024 points); however, further ablation on the S3DIS dataset (Table R1-3) validates SIDA's significant efficacy in semantic segmentation. Furthermore, by selectively deploying SIDA in later, downsampled stages (Appendix B), the method minimizes computational overhead and maintains substantial energy efficiency, as corroborated by Tables 2, 4, and 5.
>
> #### Table R1-3
> | Architecture| OA(%) | mAcc(%) | mIoU(%)|
> | -------- |:--------:|:--------:|:--------:|
> | SEDA+SIDA  | 90.1 | 76.8    |  69.6  |
> | SEDA  | 89.6  |  75.6    |  68.4   |
> | SIDA  | 89.5  |  75.3    |  68.2    |
> | SSA   | 88.9  |  74.1    |  66.9    |
>
>
> [1] Spike-driven Transformer V2: Meta Spiking Neural Network Architecture Inspiring the Design of Next-generation Neuromorphic Chips. ICLR 2024
>
> [2] SpikingResformer: Bridging ResNet and Vision Transformer in Spiking Neural Networks. CVPR 2024
>
> [3] QKFormer: Hierarchical Spiking Transformer using Q-K Attention. NIPS 2024
>
> [4] kNN-STUFF: kNN STreaming Unit for Fpgas. IEEE Access 2019

---

### Author Response · Authors · 2025-12-03
**Summary**

For the convenience of the AC, we have summarized the reviewers' feedback and our responses as follows.
### Strengths

**Methodological Innovation:** All reviewers recognize the novelty of the proposed Spiking Discrepancy Transformer (SDT) and the Spiking Discrepancy Attention Mechanism (SDAM). The integration of bio-inspired dynamics with multi-scale attention was well-received (Reviewers krS2, Aiaz, UkDr).

**Experimental Performance:** The reviewers collectively highlight the model's superior performance on classification and segmentation tasks, as well as its efficiency regarding parameter count and energy consumption.

**Presentation Quality:** Reviewers UkDr and krS2 specifically appreciate the paper’s narrative coherence, logical flow, and the clarity of the motivation presented.


### Questions
**More Experimental Results:**

The reviewers expressed concerns regarding the robustness and practical utility of our proposed model. In response, we compared our Spiking Discrepancy Attention Mechanism (SDAM) against various existing 2D spiking attention mechanisms and further validated the efficacy of SIDA on semantic segmentation datasets by benchmarking against additional state-of-the-art (SOTA) SNN models. Furthermore, we provided a clearer demonstration of the utility of the Spatially-Aware Spiking Neuron (SASN) and conducted sensitivity analyses on the scale hyperparameter within SDAM. These results collectively demonstrate the robustness and effectiveness of our individual modules as well as the overall methodology.

**Concerns on Theoretical Details:**

Reviewer krS2 inquired about the advantages of SNNs for point cloud tasks beyond energy efficiency. Through our experiments, we demonstrated that SNNs exhibit superior robustness compared to ANNs, particularly against 3D transformations and minor perturbations in point cloud scenarios.

Reviewer Aiaz further questioned the theory why the SDAM is more suitable for point clouds than original Spiking Self-Attention (SSA). In both the revised manuscript and this rebuttal, we have articulated from the perspective of information entropy. The discrepancy metric is intrinsically better suited for point cloud analysis. Besides, we demonstrated its capacity to efficiently capture both local and global 3D structures while significantly reducing computational complexity.

Reviewer UkDr's concerns regarding hardware compatibility are addressed by preprocessing coordinate operations offline and leveraging dedicated accelerators for feature-based KNNs.

---

### Meta-Review · Area_Chair_J41b · 2026-01-07

**Summary:**

This paper presents a spiking architectures for point cloud analysis, with a dual-form discrepancy-based attention mechanism and a spatially aware neuron that together deliver competitive performance with improved efficiency. The authors added substantial new theory, experiments and clarifications during rebuttal, and these fully addressed the main concerns raised across reviewers. With reviewers either maintaining (likely) or raising their scores and expressing confidence in the revised work, I recommend acceptance.

**Reviewer Concerns:**

Reviewer UkDr’s concerns about broader validation, missing comparisons and clarification regarding spike-driven operations. The rebuttal  provided new experiments across multiple attention variants, additional SNN baselines for segmentation and an explanation of how KNN and trigonometric steps can be preprocessed or offloaded to accelerators.

Reviewer krS2’s questions about the benefits of SNNs beyond energy savings, the role of temporal timesteps and unclear ablations were resolved through added robustness studies, clearer baselines and empirical demonstrations that SNNs degrade less under perturbations.

The authors addressed Reviewer Aiaz’s concerns, including an information-entropy argument contrasting self-attention similarity and spiking discrepancy, attention map comparisons, clarification of symbols and module definitions and ablations on the scaling factor.

The main point that might remain only partially resolved is UkDr’s concern that SASN and KNN prevent fully spike-native deployment, since the model still depends on non-spiking computations even if they can be offloaded; additionally, Aiaz’s request for rigorously formal theoretical justification is answered through principled arguments and supporting experiments rather than airtight mathematical proof.

**Reviewer Scores:**

For Reviewer UkDr, based on the original tone of the review, which was positive score of 6 , it is expect that the reviewer would likely maintain the same score.

For Reviewer krS2, the reviewer explicitly wrote after reading the rebuttal that they believed most concerns had been addressed and stated an intention to raise the score to 8, indicating a clear upward trend.

For Reviewer Aiaz, there is no direct follow-up comment after the rebuttal. the reviewer’s original concerns were technical questions. The authors added substantial theoretical elaboration, sensitivity studies and clearer definitions to meet those requests. The score would likely remain at 6.

---

### Decision · Program_Chairs · 2026-01-26

Accept (Poster)